# DEEP PROBABILITY ESTIMATION

## ABSTRACT

Reliable probability estimation is of crucial importance in many real-world applications where there is inherent uncertainty, such as weather forecasting, medical prognosis, or collision avoidance in autonomous vehicles. Probability-estimation models are trained on observed outcomes (e.g. whether it has rained or not, or whether a patient has died or not), because the ground-truth probabilities of the events of interest are typically unknown. The problem is therefore analogous to binary classification, with the important difference that the objective is to estimate probabilities rather than predicting the specific outcome. The goal of this work is to investigate probability estimation from high-dimensional data using deep neural networks. There exist several methods to improve the probabilities generated by these models but they mostly focus on classification problems where the probabilities are related to model uncertainty. In the case of problems with inherent uncertainty, it is challenging to evaluate performance without access to ground-truth probabilities. To address this, we build a synthetic dataset to study and compare different computable metrics. We evaluate existing methods on the synthetic data as well as on three real-world probability estimation tasks, all of which involve inherent uncertainty: precipitation forecasting from radar images, predicting cancer patient survival from histopathology images, and predicting car crashes from dashcam videos. Finally, we also propose a new method for probability estimation using neural networks, which modifies the training process to promote output probabilities that are consistent with empirical probabilities computed from the data. The method outperforms existing approaches on most metrics on the simulated as well as real-world data.

## 1 INTRODUCTION

We consider the problem of building models that answer questions such as: *Will it rain? Will a patient survive? Will a car collide with another vehicle?* Due to the inherently-uncertain nature of these real-world phenomena, this requires performing *probability estimation*, i.e. estimating the probability of each possible outcome of the phenomenon of interest. Models for probability prediction must be trained on observed outcomes (e.g. whether it rained, a patient died, or a collision occurred), because the ground-truth probabilities are unknown. The problem is therefore analogous to binary classification, with the important difference that the objective is to estimate probabilities rather than predicting specific outcomes. In probability estimation, two identical inputs (e.g. histopathology images from cancer patients) can potentially result in two different outcomes (death vs. survival). In contrast, in classification the class label is usually completely determined by the data (a picture either shows a cat or it does not).

The goal of this work is to investigate probability estimation from high-dimensional data using deep neural networks. Deep networks trained for classification often generate probabilities, which quantify the uncertainty of the estimate (i.e. how likely the network is to classify correctly). This quantification has been observed to be inaccurate, and several methods have been developed to improve it (Platt, 1999; Guo et al., 2017; Szegedy et al., 2016; Zhang et al., 2020; Thulasidasan et al., 2020; Mukhoti et al., 2020; Thagaard et al., 2020), including Bayesian neural networks (Gal & Ghahramani, 2016; Wang et al., 2016; Shekhovtsov & Flach, 2019; Postels et al., 2019). However, these works restrict their attention almost exclusively to classification in datasets (e.g. CIFAR-10/100 Krizhevsky (2009), or ImageNet (Deng et al., 2009)) where the label is *not uncertain*, and therefore the uncertainty is completely tied to the model: it quantifies the confidence of the model in its own prediction, *not the probability of an event of interest*.

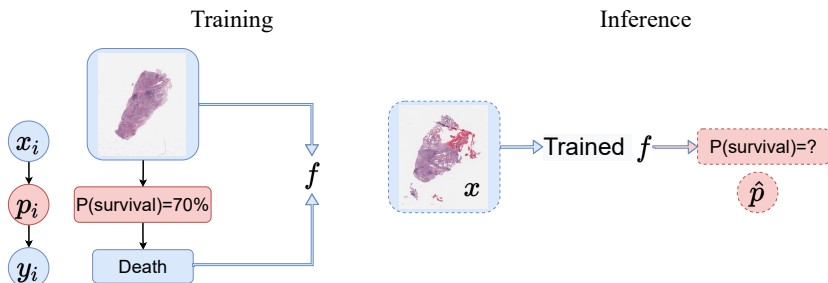

Figure 1: **The probability-estimation problem.** In probability estimation, we assume that each observed outcome $y_i$ (e.g. death or survival in cancer patients) in the training set is randomly generated from a latent unobserved probability $p_i$ associated to the corresponding data $x_i$ (e.g. histopathology images). **Training** (left): Only $x_i$ and $y_i$ can be used for training, because $p_i$ is not observed. **Inference** (right): Given new data $x$, the trained network $f$ produces a probability estimate $\hat{p} \in [0, 1]$.

Probability estimation from high-dimensional data is crucial in medical prognostics (Wulczyn et al., 2020), weather prediction (Agrawal et al., 2019), and autonomous driving (Kim et al., 2019). In order to advance deep-learning methodology for probability estimation it is crucial to build appropriate benchmark datasets. Here we build a synthetic dataset and gather three real-world datasets, which we use to systematically evaluate existing methodology. In addition, we propose a novel approach, which outperforms current state-of-the-art methods. Our contributions are the following:

- We introduce a new synthetic dataset for probability estimation where a population of people may have a certain disease connected to age. The task is to predict the probability that they contract the disease from a picture of their face. The data are generated based on the UTKFaces dataset (Zhang et al., 2017a), which contains age information. The dataset contains multiple versions of the synthetic labels, which are generated according to different distributions designed to mimic real-world probability-prediction datasets. The dataset serves two objectives. First, it allows us to evaluate existing methodology. Second, it enables us to evaluate different metrics in a controlled scenario where we have access to ground-truth probabilities.
- We have used publicly available data to build probability-estimation benchmark datasets for three real-world applications: (1) precipitation forecasting from radar images, (2) prediction of cancer-patient survival from histopathology images, and (3) prediction of vehicle collisions from dashcam videos. We use these datasets to systematically evaluate existing approaches, which have been previously tested mainly on classification datasets.
- We propose Calibrated Probability Estimation (CaPE), a novel technique which modifies the training process so that output probabilities are consistent with empirical probabilities computed from the data. CaPE outperforms existing approaches on most metrics on synthetic and real-world data.

## 2 PROBLEM FORMULATION: PROBABILITY ESTIMATION

The goal of probability estimation is to evaluate the likelihood of a certain event of interest, based on observed data. The available training data consists of $n$ examples $x_i$, $1 \le i \le n$, each associated with a corresponding outcome $y_i$. In our applications of interest, the input data are high dimensional: each $x_i$ corresponds to an image or a video. The corresponding label $y_i$ is either 0 or 1 depending on whether or not the event in question occurred. For example, in the cancer-survival application $x_i$ is a histopathology image of a patient, and $y_i$ equals 1 if the patient survived for 5 years after $x_i$ was collected. The data have inherent uncertainty: $y_i$, the patient's survival, does not depend deterministically on the histopathology image (due e.g. to comorbidities and other health factors). Instead, we assume that $y_i$ equals 1 with a certain probability $p_i$ associated with $x_i$,, as illustrated in Figure 1, because the input data provides key information about the patient's survival chances.

At inference, a probability-estimation model aims to generate an estimate $\hat{p}$ of the underlying probability $p$, associated with a new input data point $x$ (e.g. the probability of survival for over 5 years for new patients based on their histopathology data). To summarize, this is not a classification problem, because the labels are not completely predictable. Instead, the goal is to predict the probability of the outcome, which is critical in choosing a course of treatment for the patient.

(a) ✓ Accurate probability estimation
✓ Calibration

(b) ✗ Inaccurate probability estimation
✓ Calibration

Figure 2: **Calibration is not enough**. Uncolored/colored markers denote $y = 0/1$ outcomes, respectively. Blue/red stand for two classes with different associated ground-truth probabilities (1/4 and 3/4 respectively). $(a)$ The model $f$ retrieves the true probabilities, which requires discriminating between inputs with low and high probability. $(b)$ The model $f$ has no discriminative power, it just assigns the same probability to all outputs. However, the model is *perfectly calibrated* because out of all outcomes assigned 0.5 by the model, the fraction that are equal to 1 is 50%.

## 3 EVALUATION METRICS

Probability estimation shares similar target labels and network outputs with binary classification. However, classification accuracy is *not* an appropriate metric for evaluating probability-estimation models due to the inherent uncertainty of the outcomes. This is illustrated by the example in Figure 2a where a perfect probability estimate would result in a classification accuracy of just 75%.[1]

**Metrics when ground-truth probabilities are available.** For synthetic datasets, we have access to the ground truth probability labels and can use them to evaluate performance. Two reasonable metrics are the mean squared error or $\ell_2$ distance $\text{MSE}_p$, and the Kullback–Leibler divergence $\text{KL}_p$ between the estimated and ground-truth probabilities:

$$\text{MSE}_p = \frac{1}{N} \sum_{i=1}^{N} (\hat{p}_i - p_i)^2, \text{ and } \text{KL}_p = \frac{1}{N} \sum_{i=1}^{N} \left( \hat{p}_i \log\left(\frac{\hat{p}_i}{p_i}\right) + (1 - \hat{p}_i) \log\left(\frac{1 - \hat{p}_i}{1 - p_i}\right) \right). \quad (1)$$

$N$ is the number of data, and $p_i, \hat{p}_i$ are the ground-truth and predicted probabilities respectively.

**Calibration metrics.** In real-world data, ground-truth probabilities are not available. In order to evaluate the probabilities estimated by a model, we need to compare them to the observed probabilities. To this end, we aggregate the examples for which the model output equals a certain value (e.g. 0.5), and verify what fraction of them have outcomes equal to 1. If the fraction is close to the model output, then the model is said to be well calibrated.

**Definition 3.1.** A model $f$ is well *calibrated* if

$$\mathbb{P}(y = 1 \mid f(\boldsymbol{x}) \in I(q)) = q, \quad \forall \, 0 \leq q \leq 1, \quad (2)$$

where $y$ is the observed outcome, $f(\boldsymbol{x})$ is the probability predicted by model $f$ for input $\boldsymbol{x}$, and $I(q)$ is a small interval around $q$.

Model calibration can be evaluated using the expected calibration error (ECE) (Guo et al., 2017) (note however that the definition Guo et al. (2017) is specific to classification). Given a probability-estimation model $f$ and a dataset of input data $\boldsymbol{x}_i$ and associated outcomes $y_i$, $1 \leq i \leq N$, we partition the examples into $B$ bins, $I_1 I_2, \cdots, I_B$, according to the probabilities assigned to the examples by the model. Let $Q_1, \ldots, Q_{B-1}$ the $B$-quantiles of the set $\{f(\boldsymbol{x}_1), \ldots, f(\boldsymbol{x}_N)\}$, we have $I_b := [Q_{b-1}, Q_b] \cap \{f(\boldsymbol{x}_i)\}_{i=1}^N$ (setting $Q_0 = 0$). For each bin, we compute the mean predicted and empirical probabilities,

$$p_{\text{emp}}^{(b)} = \mathbb{E}(y \mid f(\boldsymbol{x}) \in I_b) = \frac{1}{|I_b|} \sum_{i \in \text{Index}(I_b)} y_i, \quad (3)$$

$$q^{(b)} = \frac{1}{|I_b|} \sum_{i \in \text{Index}(I_b)} f(\boldsymbol{x}_i), \quad (4)$$

---

[1]A perfect model (in terms of probability estimation), assigns 0.25 to the blue class and 0.75 to the red class. To maximize classification accuracy, we predict when the model outputs 0.75 (red examples) and 0 when it outputs 0.25 (blue examples). However, 25% of red examples have an outcome of 0, and 25% of blue examples have an outcome of 1. As a result, the model would only have 75% accuracy.

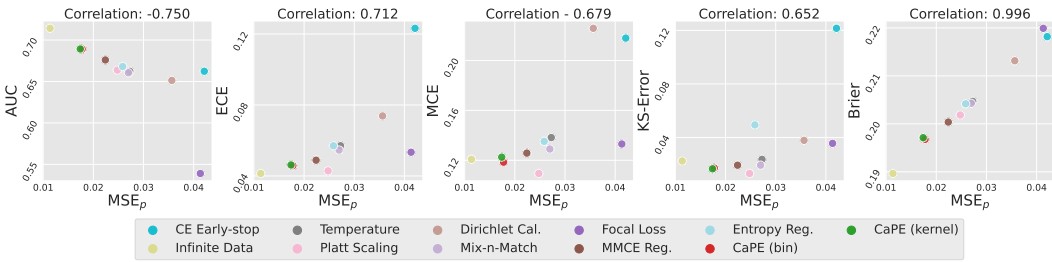

Figure 3: **Evaluating evaluation metrics.** We use synthetic data to compare different metrics to the *gold-standard* $\mathrm{MSE}_p$ that uses ground-truth probabilities. Brier score is highly correlated with $\mathrm{MSE}_p$, in contrast to the classification metric AUC and the calibration metrics ECE, MCE and KS-Error. The graphs show the results of the proposed method CaPE, as well as the baselines described in Section 6.3 on the *Linear* scenario (see Section 6.1. Other scenarios and a similar comparison with $\mathrm{KL}_p$ are included in Appendix D.)

where $\mathrm{Index}(I_b) = \{i \mid f(\boldsymbol{x}_i) \in I_b\}$.

The pairs $(q^{(b)}, p_{\mathrm{emp}}^{(b)})$ can be plotted as a reliability diagram, shown in the second row of Figure 4 and in Figure 6. ECE is then defined as

$$\mathrm{ECE} = \frac{1}{B} \sum_{b=1}^{B} \left| p_{\mathrm{emp}}^{(b)} - q^{(b)} \right|. \tag{5}$$

Other metrics for calibration include the maximum calibration error (MCE) defined as

$$\mathrm{MCE} = \max_{b=1,\dots,B} \left| p_{\mathrm{emp}}^{(b)} - q^{(b)} \right|,$$

and the Kolmogorov-Smirnov error (KS-error) (Gupta et al., 2021), a metric based on the cumulative distribution function, which is described in more detail in Appendix B.

**Brier score.** Crucially, a model without any discriminative power can be perfectly calibrated (see Figure 2b). The Brier score is a metric designed to evaluate both calibration and discriminative ability. It is the mean squared error between the predicted probability and the observed outcomes:

$$\mathrm{Brier} = \frac{1}{N} \sum_{i=1}^{N} (\hat{p}_i - y_i)^2. \tag{6}$$

This score can be decomposed into two terms associated to calibration and discrimination ability, as shown in Appendix C. Using the synthetic data in Section 6.1, where the ground-truth probabilities are known, we show that Brier score is indeed a reliable proxy for *gold-standard* MSE metric based on ground-truth probabilities $\mathrm{MSE}_p$, in contrast to calibration metrics such as ECE, MCE or KS-error, and to classification metrics such as AUC (see Figure 3 and Appendix D).

## 4 PROPOSED METHOD: CALIBRATED PROBABILITY ESTIMATION (CAPE)

Prediction models based on deep learning are typically trained by minimizing the cross entropy between the model output and the training labels (Goodfellow et al., 2016). This cost function is a *proper scoring rule*, which means that it evaluates probability estimates in a consistent manner and is therefore guaranteed to be well calibrated in an infinite-data regime (Buja et al., 2005), as illustrated by Figure 4 (first column).

Unfortunately, in practice prediction models are trained on finite data. This is crucial in the case of deep neural networks, because these models are highly overparametrized and therefore prone to overfitting (Goodfellow et al., 2016). In classification, networks have been shown to be capable of fitting arbitrary random labels (Zhang et al., 2017a). In probability estimation, we observe that neural networks indeed eventually overfit the observed outcomes completely. Moreover, the estimated probabilities collapse to 0 or 1 (Figure 4, second column), a phenomenon that has also been reported in classification (Mukhoti et al., 2020). However, calibration is preserved during the first stages of training (Figure 4, third column). This is reminiscent of the *early-learning* phenomenon observed

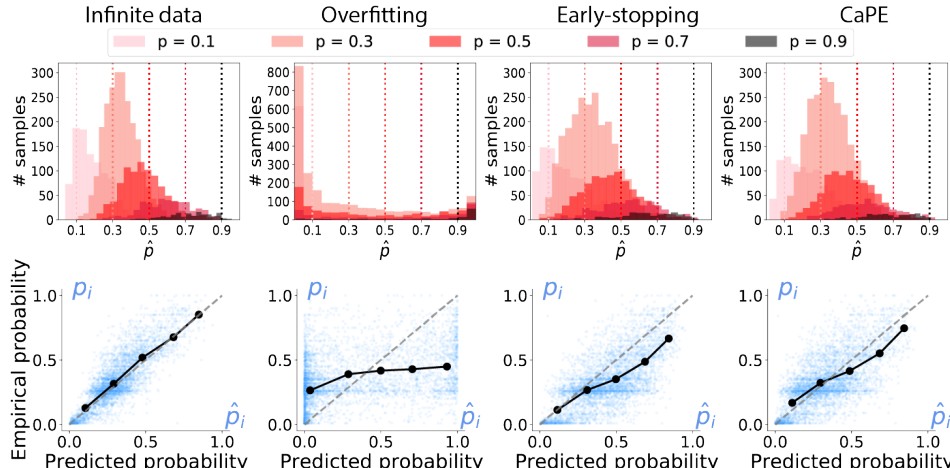

Figure 4: **Miscalibration due to overfitting and how to avoid it.** When trained on *infinite data* (i.e. resampling outcome labels at each epoch according to ground-truth probabilities), models minimizing cross-entropy are well calibrated (first column). The top row shows results for the synthetic *Discrete* scenario (see Section 6.1) (top). The bottom row shows results for the *Linear* scenario (dashed line indicates perfect calibration). However, when trained on fixed observed outcomes, the model eventually overfits and the probabilities collapse to either 0 or 1 (second column). This is mitigated via early stopping (i.e. selecting the model based on validation cross-entropy loss), which yields relatively good calibration (third column). The proposed Calibration Probability Estimation (CaPE) method exploits this to further improve the model while ensuring that the output remains well calibrated. Appendix A.3 shows plots for all synthetic data scenarios.

for classification from partially corrupted labels (Yao et al., 2020; Xia et al., 2020), where neural networks learn from the correct labels before eventually overfitting the false ones (Liu et al., 2020).

Here, we propose to exploit the training dynamics of cross-entropy minimization through a method that we name *Calibrated Probability Estimation* (CaPE). Our starting point is a model obtained via early stopping using validation data on the cross-entropy loss. CaPE is designed to further improve the discrimination ability of the model, while ensuring that it remains well calibrated. This is achieved by alternatively minimizing the following two loss functions:

**Discrimination loss**: Cross entropy between the model output and the observed binary outcomes,

$$\mathcal{L}_{\mathrm{D}} = -\sum_{i=1}^{N} \left[ y_i \log(f(\boldsymbol{x}_i)) + (1 - y_i) \log(1 - f(\boldsymbol{x}_i)) \right].$$

**Calibration loss**: Cross entropy between the output probability of the model and the empirical probability of the outcomes conditioned on the model output:

$$\mathcal{L}_{\mathrm{C}} = -\sum_{i=1}^{N} \left[ p_{\mathrm{emp}}^i \log(f(\boldsymbol{x}_i)) + (1 - p_{\mathrm{emp}}^i) \log(1 - f(\boldsymbol{x}_i)) \right],$$

where $p_{\mathrm{emp}}^i$ is an estimate of the conditional probability $\mathbb{P}[y = 1 | f(\boldsymbol{x}) \in I(f(\boldsymbol{x}_i))]$ where $I(f(\boldsymbol{x}_i))$ is a small interval centered at $f(\boldsymbol{x}_i)$. As explained in Section 3 if $f(x_i)$ is close to this value, then the model is well calibrated. We consider two approaches for estimating $p_{\mathrm{emp}}^i$. (1) CaPE (bin) where we divide the training set into bins, select the bin $b_i$ containing $f(x_i)$ and set $p_{\mathrm{emp}}^i = p_{\mathrm{emp}}^{(b_i)}$ in equation 3. (2) CaPE (kernel) where $p_{\mathrm{emp}}^i$ is estimated through a moving average with a kernel function (see Appendix E for more details). Both methods are efficiently computed by sorting the predictions $\hat{p}_i$. The calibration loss requires a reasonable estimation of the empirical probabilities $p_{\mathrm{emp}}^{(i)}$, which can be obtained from the model after early learning. Therefore using the calibration loss from the beginning is counterproductive, as demonstrated in Section J.

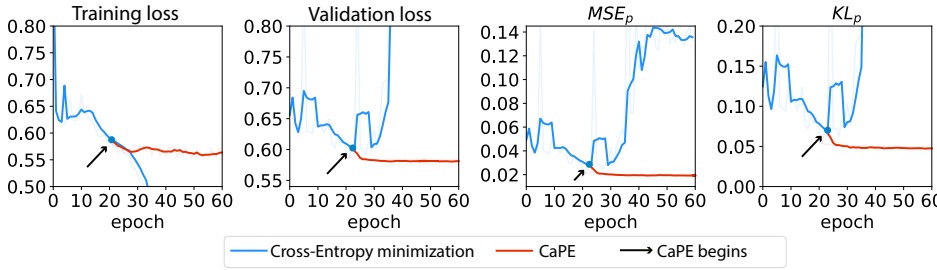

Figure 5: **Calibrated Probability Estimation prevents overfitting.** Comparison between the learning curves of cross-entropy (CE) minimization and the proposed calibrated probability estimation (CaPE), smoothed with a 5-epoch moving average. After an *early-learning* stage where both training and validation losses decrease, CE minimization overfits (first and second graph), with disastrous consequences in terms of probability estimation (third and fourth graph). In contrast, CaPE prevents overfitting, continuing to improve the model while maintaining calibration (see Figure 4).

---

**Algorithm 1** Pseudocode for CaPE

---

**Require:** $f$            ▷ early stopped model
**Require:** $m$          ▷ freq. of training with $\mathcal{L}_C$
**Require:** $\{\boldsymbol{x}_i, y_i\}_{i=1}^N$      ▷ training set
**Require:** $K(p, q) \coloneqq \exp\left[-(p-q)^2/\sigma^2\right]$ ▷ Gaussian kernel
  **for** $t = 1$ to `num_epochs` **do**
    **if** $t \mod m = 0$ **then**
      $\hat{p}_i \leftarrow f(\boldsymbol{x}_i), \forall i$
      Update $p_{\text{emp}}^i, \forall i$, with BIN or KERNEL
      $\mathcal{L} \leftarrow \mathcal{L}_C$     ▷ compute discrimination loss
    **else**
      $\mathcal{L} \leftarrow \mathcal{L}_D$        ▷ compute calibration loss
    **end if**
    $f \leftarrow$ backprop with $\mathcal{L}$     ▷ train with loss
  **end for**

**function** BIN($B$)         ▷ $B$-number of bins
  $I_1, \cdots I_B \leftarrow$ partitions by quantile of $\{\hat{p}_j\}_{j=1}^N$
  Find $b$ such that $\hat{p}_i \in I_b$
  Index$(I_b) \leftarrow \{j | \hat{p}_j \in I_b\}$    ▷ get indices in bin $b$
  $p_{\text{emp}}^i \leftarrow \frac{1}{|I_b|} \sum_{i \in \text{Index}(I_b)} y_i$    ▷ empirical mean of bin $b$
**end function**

**function** KERNEL($r, K$)     ▷ $r$-window size; kernel
  $N_r(i) \leftarrow r$-nearest neighbor of $\hat{p}_i$ (output probability space)
  $Z \leftarrow \sum_{\hat{p}_j \in N_r(i)} K(\hat{p}_i, \hat{p}_j)$    ▷ normalization factor
  $p_{\text{emp}}^i \leftarrow \sum_{\hat{p}_j \in N_r(i)} K(\hat{p}_i, \hat{p}_j)\, y_j / Z$    ▷ kernel smooth
**end function**

---

CaPE is summarized in Algorithm 1. Figures 4 and 5 show that incorporating the calibration-loss minimization step indeed preserves calibration as training proceeds (this is not necessarily expected because CaPE minimizes a calibration loss *on the training data*), and prevents the model from overfitting the observed outputs. This is beneficial also for the discriminative ability of the model, because it enables it to further reduce the cross-entropy loss without overfitting, as shown in Figure 5. The experiments with synthetic and real-world data reported in Section 6 suggest that this approach results in accurate probability estimates across a variety of realistic scenarios.

## 5 RELATED WORK

Neural networks trained for classification often generate a probability associated with their prediction which quantifies its uncertainty. These estimates are often found to be inaccurate (Mukhoti et al., 2020; Guo et al., 2017). Techniques mitigating this issue are often described as calibration methods, and broadly fall into three categories depending on whether they: (1) postprocess the outputs of a trained model, (2) combine multiple model outputs, or (3) modify the training process.

**Post-processing methods** transform the output probabilities in order to improve calibration on held-out data (Zadrozny & Elkan, 2001; Gupta et al., 2021; Kull et al., 2017; 2019). For example, Platt scaling (Platt, 1999) fits a logistic function that minimizes the negative log-likelihood loss. Temperature scaling (Guo et al., 2017) does the same with a temperature parameter augmenting the softmax function. In contrast to these methods, CaPE enforces calibration *during training*, which has the advantage of enabling further improvements in the discriminative abilities of the model.

**Ensembling methods** combine multiple models to improve generalization. Mix-n-Match (Zhang et al., 2020) uses a single model, and ensembles predictions using multiple temperature scaling transformations. Other methods (Lakshminarayanan et al., 2017; Maddox et al., 2019) ensemble

| Methods | Linear | | Sigmoid | | Centered | | Skewed | | Discrete | |
|---|---|---|---|---|---|---|---|---|---|---|
| $(\times 10^{-2})$ | $\text{MSE}_p$ | $\text{KL}_p$ | $\text{MSE}_p$ | $\text{KL}_p$ | $\text{MSE}_p$ | $\text{KL}_p$ | $\text{MSE}_p$ | $\text{KL}_p$ | $\text{MSE}_p$ | $\text{KL}_p$ |
| Infinite Data* | 1.13 | 2.81 | 5.35 | 14.86 | 0.20 | 0.41 | 0.22 | 0.92 | 1.52 | 3.64 |
| CE early-stop | 4.21 | 10.93 | 6.16 | 17.16 | 0.48 | 0.98 | 0.40 | 1.79 | 2.24 | 5.26 |
| Temperature | 2.71 | 6.70 | 6.13 | 17.05 | 0.48 | 0.98 | 0.40 | 1.75 | 2.20 | 5.12 |
| Platt Scaling | 2.48 | 6.06 | 5.78 | 16.12 | 0.41 | 0.83 | **0.39** | 1.71 | 2.06 | 4.82 |
| Dirichlet Cal. | 3.56 | 9.06 | 8.71 | 25.33 | 0.46 | 0.94 | 0.48 | 2.31 | 2.74 | 6.52 |
| Mix-n-match | 2.69 | 6.70 | 6.13 | 17.10 | 0.48 | 0.98 | **0.39** | 1.74 | 2.20 | 5.12 |
| Focal Loss | 4.13 | 10.51 | 6.89 | 19.51 | 0.48 | 0.97 | 1.27 | 11.61 | 2.92 | 6.76 |
| Entropy Reg. | 2.84 | 7.37 | 7.03 | 21.19 | 0.42 | 0.87 | 1.17 | 10.57 | 2.84 | 6.62 |
| MMCE Reg. | 2.22 | 5.65 | 5.33 | 15.03 | 0.44 | 0.90 | 0.54 | 2.43 | 2.08 | 4.90 |
| Deep Ensemble | 1.90 | 4.55 | 5.85 | 16.43 | 0.44 | 0.89 | 0.55 | 2.58 | 1.97 | 4.61 |
| CaPE (bin) | 1.83 | 4.46 | 5.29 | 14.59 | **0.38** | **0.78** | 0.40 | 1.72 | **1.83** | **4.31** |
| CaPE (kernel) | **1.81** | **4.41** | **5.22** | **14.47** | 0.40 | 0.81 | **0.39** | **1.70** | 1.85 | 4.36 |

Table 1: Results on synthetic data. Appendix A.1 shows a table with confidence intervals using bootstrapping. * is a model trained with infinite data obtained by continuous label resampling.

multiple models obtained using different initializations. These approaches are compatible with the proposed method CaPE; how to combine them effectively is an interesting future research direction.

**Modified training methods** can be divided into two groups. The first group smooths the target 0/1 labels in order to prevent output estimates from collapsing to 0/1 (Mukhoti et al., 2020; Szegedy et al., 2016; Zhang et al., 2018; Thulasidasan et al., 2020). The second group, attaches additional calibration penalties to a cross entropy loss (Kumar et al., 2018; Pereyra et al., 2017; Liang et al., 2020). CaPE is most similar in spirit to the latter methods, although its data-driven calibration loss is different to the penalties used in these techniques.

**Datasets for evaluation** The methods discussed in this section were developed for calibration in classification, and tested on datasets such as CIFAR-10/100 (Krizhevsky, 2009), SVHN (Netzer et al., 2011), and ImageNet (Deng et al., 2009) where the relationship between labels and input data is completely deterministic. Here, we evaluate these methods on synthetic and real-world probability-estimation problems with inherent uncertainty.

# 6 EXPERIMENTS

## 6.1 SYNTHETIC DATASET: FACE-BASED RISK PREDICTION

To benchmark probability-estimation methods, we build a synthetic dataset based on UTK-Face (Zhang et al., 2017b), containing face images and associated ages. We use the age of the $i$th person $z_i$ to assign them a risk of contracting a disease $p_i = \psi(z_i)$ for a fixed function $\psi : \mathbb{N} \to [0, 1]$. Then we simulate whether the person actually contracts the illness (label $y_i = 1$) or not ($y_i = 0$) with probability $p_i$. The probability-estimation task is to estimate the ground-truth probability $p_i$ from the face image $x_i$, which requires learning to discriminate age and map it to the corresponding risk. We design $\psi$ to create five scenarios, inspired by real-world data (see Appendix G):

- **Linear**: Equally-spaced, inspired by weather forecasting: $\psi(z) = z/100$
- **Sigmoid**: Concentrated near two extremes: $\psi(z) = \sigma(25(z/100 - 0.29))$
- **Skewed**: Clustered close to zero, inspired by vehicle-collision detection: $\psi(z) = z/250$
- **Centered**: Clustered in the center, inspired by cancer-survival prediction: $\psi(z) = z/300 + 0.35$
- **Discrete**: Discretized: $\psi(z) = 0.2 \left[ \mathbb{1}_{\{z>20\}} + \mathbb{1}_{\{z>40\}} + \mathbb{1}_{\{z>60\}} + \mathbb{1}_{\{z>80\}} \right] + 0.1$

## 6.2 REAL-WORLD DATASETS

We propose to use three open-source, real-world datasets to benchmark probability-estimation approaches (see Appendix H for further details on the datasets and experiments).

| Method | Cancer Survival | | | Weather forecasting | | | Collision Prediction | | |
|---|---|---|---|---|---|---|---|---|---|
| ($\times 10^{-2}$) | AUC | ECE | Brier | AUC | ECE | Brier | AUC | ECE | Brier |
| CE early-stop | 58.88 | 12.25 | 23.96 | 77.64 | 10.91 | 20.57 | 85.68 | 4.36 | 8.59 |
| Temperature | 58.88 | 12.07 | 23.73 | 77.64 | 8.66 | 20.21 | 85.68 | 4.56 | 8.51 |
| Platt Scaling | 58.91 | 10.28 | 23.33 | 77.65 | 6.97 | 19.53 | 85.76 | 3.04 | 8.23 |
| Dirichlet Cal. | 49.89 | 13.83 | 24.08 | 77.51 | 14.29 | 21.89 | 83.36 | 5.78 | 8.78 |
| Mix-n-match | 58.88 | 12.16 | 23.67 | 77.64 | 8.65 | 20.21 | 85.68 | 4.40 | 8.52 |
| Focal Loss | 55.02 | 12.15 | 23.31 | 76.18 | 8.32 | 20.27 | 82.21 | 9.07 | 9.82 |
| Entropy Reg. | 56.29 | 11.73 | 23.62 | 79.01 | 10.53 | 19.77 | 83.15 | 14.54 | 11.10 |
| MMCE Reg. | 48.45 | 11.84 | 23.73 | 76.69 | 8.46 | 20.12 | 85.18 | **2.94** | 8.48 |
| Deep Ensemble | 52.46 | 9.99 | 23.47 | **79.86** | 7.41 | 18.82 | 85.27 | 3.15 | 8.55 |
| CaPE (bin) | **61.44** | 12.31 | 23.20 | 78.99 | 5.16 | **18.37** | 85.70 | 3.16 | 8.18 |
| CaPE (kernel) | 61.22 | **9.48** | **23.18** | 79.00 | **5.08** | 18.39 | **85.95** | 3.22 | **8.13** |

Table 2: Results on cancer-survival prediction, weather forecasting, and collision prediction. Tables with all the metrics described in Section 3 are provided in Appendix A.2

**Survival of Cancer Patients.** Histopathology aims to identify tumor cells, cancer subtypes, and the stage and level of differentiation of cancer. Hematoxylin and Eosin (H&E)-stained slides are the most common type of histopathology data used for clinical decision making. In particular, they can be used for survival prediction (Wulczyn et al., 2020), which is critical in evaluating the prognosis of patients. Treatments assigned to patients after diagnosis are not personalized and their impact on cancer trajectory is complex, so the survival status of a patient is not deterministic. In this work, we use the H&E slides of non-small cell lung cancers from The Cancer Genome Atlas Program (TCGA)[2] to estimate the the 5-year survival probability of cancer patients. The outcome distribution is similar to the *Centered* scenario in our synthetic data.

**Weather Forecasting.** The atmosphere is governed by nonlinear dynamics, hence weather forecast models possess inherent uncertainties (Richardson, 2007). Nowcasting, weather prediction in the near future, is of great operational significance, especially with increasing number of extreme inclement weather conditions (Agrawal et al., 2019; Ravuri et al., 2021). We use the German Weather service dataset[3], which contains quality-controlled rainfall-depth composites from 17 operational Doppler radars. We use 30 minutes of precipitation data to predict if the mean precipitation will increase or decrease after one hour. Three precipitation maps from the past 30 minutes serve as an input. The outcome distribution is similar to the *Linear* scenario in our synthetic data.

**Collision Prediction.** Vehicle collision is one of the leading causes of death in the world. Reliable collision prediction systems are therefore instrumental in saving human lives. These systems predict potential collisions from dashcam cameras. Collisions are influenced by many unknown factors, and hence are not deterministic. Following Kim et al. (2019), we use $0.3$ seconds of real dashcam videos from `YouTubeCrash` dataset as input, and predict the probability of a collision in the next 2 seconds. The data are very imbalanced as the number of collisions is very low, so the outcome distribution is similar to the *Skewed* scenario in our synthetic data.

## 6.3 BASELINES

We apply existing calibration methods developed for classification to probability estimation (as well as cross-entropy minimization with early-stopping): (1) **Three post-processing methods**: Temperature Scaling (Guo et al., 2017), Platt Scaling (Platt, 1999), and Dirichlet Calibration (Kull et al., 2019) applied to the best CE model, (2) **Two Ensemble Methods**: Mix-n-Match (Zhang et al., 2020) applied to best CE model, and Deep Ensemble (Lakshminarayanan et al., 2017) with 5 networks, and (3) **Three Modified Training methods**: focal loss (Mukhoti et al., 2020), entropy-maximizing loss (Pereyra et al., 2017), and MMCE regularization (Kumar et al., 2018). Appendix F provides a detailed description. For our experiments on synthetic data, we also compare against a model trained on an **infinite amount of data** by repeatedly sampling new outcomes from the ground-truth probabilities at each epoch. This provides a best-case reference for each scenario.

---

[2] https://www.cancer.gov/tcga
[3] https://opendata.dwd.de/weather/radar/

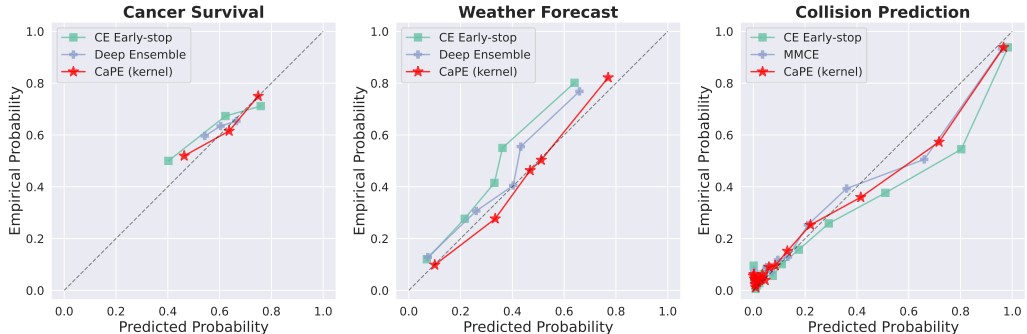

Figure 6: **Reliability diagrams for real-world data.** Reliability diagrams computed on test data for cross-entropy minimization with early stopping, the proposed method (CaPE) and the best baseline for each dataset. CaPE produces better calibrated outputs. Appendix A.3 shows additional diagrams.

## 7    RESULTS AND DISCUSSION

Table 1 shows that calibration methods developed for classification can be effective for probability estimation. However, the performance of some methods is not consistent across all scenarios. For instance, regularization with negative entropy, which penalizes very high/low confidence, performs worse than CE when the ground-truth probability is close to 0 or 1. In contrast, methods that do not make strong assumptions tend to generalize better to multiple scenarios (e.g. Platt scaling consistently beats CE). The proposed method CaPE outperforms other techniques in most scenarios, and even matches the performance of the *infinite data* baseline for the *Sigmoid* scenario. Finally, we observe that the *Skewed* scenario is very challenging: most methods barely improve the CE baseline.

Table 2 compares the baseline methods and CaPE on the three real-world datasets. We present AUC, ECE for 15 equally-sized bins, and Brier score, as complementary metrics since the underlying ground-truth probabilities are unobserved. As discussed in Section 3, Brier score is the metric that best captures the quality of probability estimates. CaPE has the lowest Brier score in all three datasets, while also achieving lower ECE values and higher AUC values than the other methods. This demonstrates that enforcing calibration during training also yields a more discriminative model. The reliability diagrams in Figure 6 depict the probability estimates produced by CE, CaPE and the best baseline method on the three datasets, demonstrating that CaPE produces a well-calibrated outputs.

Figure 6 also shows that each real-world dataset closely aligns with a particular synthetic scenario: cancer survival with *Centered*; weather forecasting with *Linear*; collision prediction with *Skewed*. This supports the significance of our synthetic benchmark dataset, and provides insights in the differences among baseline models. For example, model averaging with deep ensemble performs well on weather forecasting but has higher Brier scores than Platt scaling on the other two datasets (see Appendix I for further analysis based on pathological stages). Accordingly, deep ensemble also underperforms in the synthetic scenarios where ground-truth probabilities are clustered closely (*Sigmoid*, *Linear*), but is effective for *Linear*. Finally, as in the synthetic *Skewed* scenario, all methods had similar performance on the collision prediction task. This highlights the importance of considering different scenarios when evaluating methodology for probability estimation.

## 8    CONCLUSION

In this work we evaluate existing approaches to improve the output probabilities of neural networks on probability-estimation problems. To this end, we introduce a new synthetic benchmark dataset designed to reproduce several realistic scenarios, and also gather three real-world datasets relevant to medicine, climatology, and self-driving cars. In addition, we propose a novel approach for probability-estimation via deep learning that outperforms existing approaches for most datasets. An important application for probability estimation is in the context of survival analysis, which can be recast as estimation of conditional probabilities  (Lee et al., 2018; Shamout et al., 2020; Goldstein et al., 2021). An interesting research direction is to consider problems with several possible uncertain outcomes (analogous to multiclass classification).

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

## A  ADDITIONAL RESULTS

We present here supplementary results to the ones presented in Section 7.

### A.1  FACE-BASED RISK PREDICTION

Full evaluation with confidence intervals derived using 1000 bootstraps for the five simulated scenarios are examined: *Linear* (Table.3); *Sigmoid* (Table.4); *Centered* (Table.5); *Skewed* (Table.6); *Discrete* (Table.7). Note that all numbers are downscaled by $10^{-2}$ in the tables.

| *Linear* | ECE | MCE | KS | Brier | $MSE_p$ | $KL_p$ |
|---|---|---|---|---|---|---|
| Infinite Data | 4.14±0.81 | 12.07±3.29 | 2.24±0.88 | 18.97±0.33 | 1.14±0.04 | 2.82±0.11 |
| CE early-stop | 12.32±0.83 | 21.79±1.97 | 12.16±0.83 | 21.82±0.51 | 4.21±0.15 | 10.94±0.36 |
| Temperature | 5.7±0.74 | 13.82±2.71 | 2.36±0.74 | 20.47±0.37 | 2.73±0.11 | 6.75±0.25 |
| Platt Scaling | 4.29±0.77 | 10.94±2.55 | 1.3±0.45 | 20.18±0.36 | 2.48±0.09 | 6.07±0.22 |
| Dirichlet Cal. | 7.38±1.12 | 22.58±7.24 | 3.78±0.46 | 21.32±0.33 | 3.56±0.13 | 9.08±0.29 |
| Focal Loss | 5.34±0.68 | 13.31±2.67 | 3.56±0.85 | 21.99±0.28 | 4.13±0.11 | 10.52±0.28 |
| Mix-n-match | 5.46±0.84 | 12.9±2.51 | 1.92±0.44 | 20.43±0.35 | 2.7±0.11 | 6.72±0.24 |
| Entropy Reg. | 5.7±0.74 | 13.52±1.67 | 4.94±0.9 | 20.42±0.3 | 2.58±0.09 | 6.65±0.21 |
| MMCE Reg. | 4.89±0.74 | 12.57±2.27 | 1.92±0.46 | 20.04±0.38 | 2.24±0.08 | 5.68±0.2 |
| Deep Ensemble | 4.26±0.72 | 11.33±2.38 | 1.95±0.61 | 19.88±0.32 | 1.9±0.07 | 4.55±0.18 |
| CaPE (bin) | 4.58±0.75 | 11.85±2.49 | 1.71±0.51 | 19.68±0.36 | 1.78±0.07 | 4.35±0.16 |
| CaPE (kernel) | 4.62±0.62 | 12.25±2.31 | 1.65±0.38 | 19.71±0.34 | 1.74±0.07 | 4.3±0.17 |

Table 3: Performance on Face-based Risk Prediction. *Linear* scenario.

| *Sigmoid* | ECE | MCE | KS | Brier | $MSE_p$ | $KL_p$ |
|---|---|---|---|---|---|---|
| Infinite Data | 6.4±0.71 | 20.63±3.44 | 2.74±0.45 | 16.28±0.44 | 5.34±0.2 | 14.82±0.51 |
| CE early-stop | 6.19±0.75 | 17.0±3.68 | 5.86±0.8 | 16.68±0.42 | 6.16±0.17 | 17.16±0.48 |
| Temperature | 5.57±0.71 | 15.32±3.09 | 5.02±0.83 | 16.58±0.34 | 6.13±0.17 | 17.09±0.43 |
| Platt Scaling | 3.45±0.68 | 10.32±2.79 | 1.3±0.43 | 16.33±0.34 | 5.78±0.19 | 16.15±0.47 |
| Dirichlet Cal. | 14.5±1.15 | 25.68±3.02 | 4.67±0.32 | 19.21±0.43 | 8.64±0.26 | 25.18±0.58 |
| Focal Loss | 4.65±0.7 | 11.84±2.78 | 2.66±0.77 | 16.96±0.34 | 6.86±0.21 | 19.46±0.5 |
| Mix-n-match | 5.65±0.76 | 15.32±3.41 | 5.09±0.94 | 16.6±0.36 | 6.12±0.17 | 17.08±0.46 |
| Entropy Reg. | 9.51±0.79 | 18.77±2.38 | 7.26±0.78 | 17.17±0.31 | 7.02±0.17 | 21.16±0.42 |
| MMCE Reg. | 4.67±0.76 | 13.63±2.59 | 2.5±0.53 | 15.9±0.51 | 5.35±0.18 | 15.06±0.49 |
| Deep Ensemble | 5.17±0.74 | 16.12±3.11 | 2.04±0.44 | 16.39±0.45 | 5.86±0.22 | 16.46±0.6 |
| CaPE (bin) | 3.78±0.6 | 11.96±2.59 | 2.22±0.7 | 15.84±0.43 | 5.17±0.2 | 14.27±0.49 |
| CaPE (kernel) | 3.9±0.75 | 11.73±2.79 | 2.05±0.54 | 15.85±0.41 | 5.16±0.2 | 14.34±0.49 |

Table 4: Performance on Face-based Risk Prediction. *Sigmoid* scenario.

| Centered | ECE | MCE | KS | Brier | $MSE_p$ | $KL_p$ |
|---|---|---|---|---|---|---|
| Infinite Data | 4.29±0.74 | 12.38±2.92 | 2.68±0.8 | 24.22±0.13 | 0.2±0.01 | 0.41±0.01 |
| CE early-stop | 5.76±0.84 | 15.32±3.07 | 4.19±1.02 | 24.68±0.08 | 0.48±0.01 | 0.98±0.03 |
| Temperature | 6.09±0.82 | 15.83±2.91 | 4.74±0.96 | 24.74±0.06 | 0.48±0.01 | 0.98±0.03 |
| Platt Scaling | 4.57±0.76 | 11.85±2.5 | 2.79±0.85 | 24.62±0.08 | 0.41±0.01 | 0.83±0.03 |
| Dirichlet Cal. | 4.84±1.15 | 13.13±7.61 | 2.16±0.86 | 24.7±0.1 | 0.46±0.01 | 0.94±0.03 |
| Mix-n-match | 6.05±0.83 | 15.71±2.92 | 4.68±0.98 | 24.74±0.06 | 0.48±0.01 | 0.98±0.02 |
| Focal Loss | 5.09±0.83 | 13.4±2.87 | 3.44±1.02 | 24.8±0.05 | 0.48±0.01 | 0.97±0.03 |
| Entropy Reg. | 5.02±0.86 | 12.69±3.42 | 3.27±0.96 | 24.74±0.06 | 0.45±0.01 | 0.92±0.03 |
| MMCE Reg. | 5.56±0.86 | 13.59±2.65 | 2.71±0.93 | 24.7±0.08 | 0.44±0.01 | 0.9±0.03 |
| Deep Ensemble | 4.84±0.78 | 12.39±2.52 | 2.64±0.71 | 24.69±0.07 | 0.44±0.01 | 0.89±0.03 |
| CaPE (bin) | 4.73±0.82 | 11.81±2.54 | 2.07±0.6 | 24.56±0.11 | 0.38±0.01 | 0.78±0.03 |
| CaPE (kernel) | 5.41±0.87 | 12.71±2.5 | 2.39±0.78 | 24.59±0.11 | 0.4±0.01 | 0.81±0.03 |

Table 5: Performance on Face-based Risk Prediction. *Centered* scenario.

| Skewed | ECE | MCE | KS | Brier | $MSE_p$ | $KL_p$ |
|---|---|---|---|---|---|---|
| Infinite Data | 2.7±0.46 | 7.64±2.14 | 1.05±0.39 | 11.0±0.51 | 0.22±0.01 | 0.92±0.03 |
| CE early-stop | 3.07±0.57 | 7.88±1.88 | 1.28±0.41 | 11.18±0.5 | 0.4±0.01 | 1.79±0.06 |
| Temperature | 3.14±0.49 | 7.92±1.84 | 1.12±0.33 | 11.22±0.47 | 0.4±0.02 | 1.76±0.06 |
| Platt Scaling | 2.99±0.53 | 7.73±1.59 | 1.07±0.37 | 11.1±0.54 | 0.39±0.01 | 1.72±0.06 |
| Dirichlet Cal. | 3.04±0.73 | 7.81±2.43 | 0.97±0.3 | 11.22±0.42 | 0.47±0.02 | 2.31±0.07 |
| Focal Loss | 8.29±0.67 | 14.93±1.43 | 6.16±0.67 | 12.01±0.41 | 1.28±0.03 | 1.63±0.66 |
| Mix-n-match | 2.99±0.53 | 7.78±1.78 | 1.08±0.32 | 11.18±0.49 | 0.4±0.01 | 1.75±0.05 |
| Entropy Reg. | 7.67±0.57 | 14.43±1.5 | 5.2±0.71 | 11.94±0.45 | 1.18±0.03 | 10.74±0.65 |
| MMCE Reg. | 3.68±0.59 | 10.94±2.76 | 1.47±0.31 | 11.14±0.44 | 0.54±0.02 | 2.44±0.08 |
| Deep Ensemble | 2.87±0.5 | 7.21±1.63 | 1.36±0.44 | 11.28±0.5 | 0.55±0.02 | 2.58±0.07 |
| CaPE (bin) | 3.29±0.5 | 8.18±1.51 | 1.17±0.34 | 11.07±0.47 | 0.4±0.02 | 1.73±0.06 |
| CaPE (kernel) | 3.16±0.5 | 8.14±1.58 | 1.09±0.33 | 11.17±0.53 | 0.39±0.01 | 1.69±0.06 |

Table 6: Performance on Face-based Risk Prediction. *Skewed* scenario.

| Discrete | ECE | MCE | KS | Brier | $MSE_p$ | $KL_p$ |
|---|---|---|---|---|---|---|
| Infinite Data | 4.23±0.74 | 11.16±2.5 | 1.45±0.49 | 20.38±0.35 | 1.52±0.05 | 3.63±0.12 |
| CE early-stop | 6.7±0.86 | 18.62±3.52 | 2.61±0.53 | 21.91±0.36 | 2.24±0.08 | 5.27±0.17 |
| Temperature | 6.12±0.87 | 16.82±3.56 | 3.37±0.86 | 21.76±0.35 | 2.21±0.08 | 5.15±0.18 |
| Platt Scaling | 4.7±0.72 | 11.69±2.44 | 1.67±0.51 | 21.44±0.32 | 2.06±0.08 | 4.83±0.17 |
| Dirichlet Cal. | 7.13±0.86 | 22.67±5.08 | 3.18±0.68 | 22.1±0.34 | 2.74±0.1 | 6.53±0.22 |
| Focal Loss | 5.7±0.75 | 13.68±2.32 | 4.62±0.91 | 21.77±0.28 | 2.92±0.09 | 6.77±0.21 |
| Mix-n-match | 6.27±0.76 | 16.83±2.95 | 3.47±0.93 | 21.77±0.33 | 2.21±0.08 | 5.14±0.18 |
| Entropy Reg. | 6.69±0.87 | 15.38±2.43 | 6.03±1.13 | 21.79±0.31 | 2.84±0.08 | 6.62±0.19 |
| MMCE Reg. | 3.96±0.7 | 10.4±2.4 | 1.51±0.47 | 21.12±0.35 | 2.09±0.08 | 4.92±0.18 |
| Deep Ensemble | 4.76±0.74 | 11.49±2.23 | 2.04±0.61 | 21.17±0.31 | 1.97±0.08 | 4.61±0.17 |
| CaPE (bin) | 5.41±0.74 | 14.45±3.15 | 2.24±0.59 | 21.33±0.36 | 1.81±0.08 | 4.28±0.18 |
| CaPE (kernel) | 4.96±0.8 | 12.97±2.63 | 2.18±0.58 | 21.21±0.42 | 1.84±0.08 | 4.35±0.17 |

Table 7: Performance on Face-based Risk Prediction. *Discrete* scenario.

## A.2 Supplementary Metrics on Real-world Dataset

We present here additional metrics on the real world data: Cancer Survival (Table 8); Climate Forecasting (Table 9); Collision Prediction (Table 10).

| Methods ($\times 10^{-2}$) | AUC | ECE | MCE | NLL | Brier | KS |
|---|---|---|---|---|---|---|
| CE Early-stop | 58.88 | 12.25 | 25.35 | 67.92 | 23.96 | 6.44 |
| Temperature | 58.88 | 12.07 | 24.65 | 67.11 | 23.73 | 6.92 |
| Platt Scaling | 58.91 | 10.28 | 27.69 | 66.11 | 23.33 | 4.91 |
| Dirichlet Cal. | 49.89 | 13.83 | 35.52 | 67.57 | 24.08 | 6.00 |
| Mix-n-match | 58.88 | 12.16 | **24.52** | 66.89 | 23.67 | 7.18 |
| Focal loss | 55.02 | 12.15 | 26.34 | 65.92 | 23.31 | 6.38 |
| Entropy Reg. | 56.29 | 11.73 | 30.81 | 66.49 | 23.62 | 6.83 |
| MMCE Reg. | 48.45 | 11.84 | 37.36 | 66.83 | 23.73 | 3.64 |
| Deep Ensemble | 52.26 | 9.99 | 28.30 | 66.22 | 23.47 | 5.02 |
| CaPE (bin) | **61.44** | 12.31 | 25.27 | 65.75 | 23.20 | **2.59** |
| CaPE (kernel) | 61.22 | **9.48** | 32.40 | **65.70** | **23.18** | 3.70 |

Table 8: Baselines with full metrics for cancer survival

| Methods ($\times 10^{-2}$) | AUC | ECE | MCE | NLL | Brier | KS |
|---|---|---|---|---|---|---|
| CE Early-stop | 77.64 | 10.91 | 25.50 | 59.97 | 20.57 | 11.03 |
| Temperature | 77.64 | 8.66 | 23.56 | 58.77 | 20.21 | 7.41 |
| Platt Scaling | 77.65 | 6.97 | 16.47 | 57.38 | 19.53 | 3.26 |
| Dirichlet Cal. | 77.51 | 14.29 | 30.09 | 62.83 | 21.89 | 5.21 |
| Mix-n-match | 77.64 | 8.65 | 23.58 | 58.77 | 20.21 | 7.39 |
| Focal Loss | 76.18 | 8.32 | 21.25 | 59.01 | 20.27 | 4.45 |
| Entropy Reg | 79.01 | 10.53 | 20.72 | 57.83 | 19.77 | 5.00 |
| MMCE Reg | 76.69 | 8.46 | 19.73 | 59.25 | 20.12 | 7.31 |
| Deep Ensemble | **79.86** | 7.41 | 18.24 | 55.28 | 18.82 | 7.57 |
| CaPE (bin) | 78.99 | 5.16 | 15.09 | **79.00** | 18.37 | **2.34** |
| CaPE (kernel) | 79.00 | **5.08** | **13.28** | 54.32 | 18.39 | **2.34** |

Table 9: Baselines with full metrics for weather prediction

| Methods ($\times 10^{-2}$) | AUC | ECE | MCE | NLL | Brier | KS |
|---|---|---|---|---|---|---|
| CE Early-stop | 85.68 | 4.36 | 19.87 | 31.67 | 8.59 | 1.54 |
| Temperature | 85.68 | 4.56 | 16.79 | 30.36 | 8.52 | 2.9 |
| Platt Scaling | 85.76 | 3.04 | 12.39 | **29.42** | 8.23 | **1.52** |
| Dirichlet Cal. | 83.36 | 5.78 | 18.13 | 30.90 | 8.77 | 1.60 |
| Mix-n-match | 85.68 | 4.40 | 17.41 | 30.25 | 8.52 | 2.60 |
| Focal Loss | 82.21 | 9.07 | 19.85 | 34.41 | 9.82 | 8.72 |
| Entropy Reg | 83.15 | 14.54 | 21.27 | 38.74 | 11.10 | 13.44 |
| MMCE Reg. | 85.18 | **2.94** | **8.95** | 30.65 | 8.48 | 2.44 |
| Deep Ensemble | 85.27 | 3.15 | 16.53 | 30.20 | 8.54 | 2.01 |
| CaPE (bin) | 8.57 | 3.16 | 12.21 | 30.61 | 8.18 | 2.13 |
| CaPE (kernel) | **85.95** | 3.22 | 13.32 | 30.44 | **8.13** | 2.10 |

Table 10: Baselines with full metrics for collision prediction

## A.3 ADDITIONAL RELIABILITY DIAGRAM

We present here additional reliability curves to the ones illustrated in Figure 6

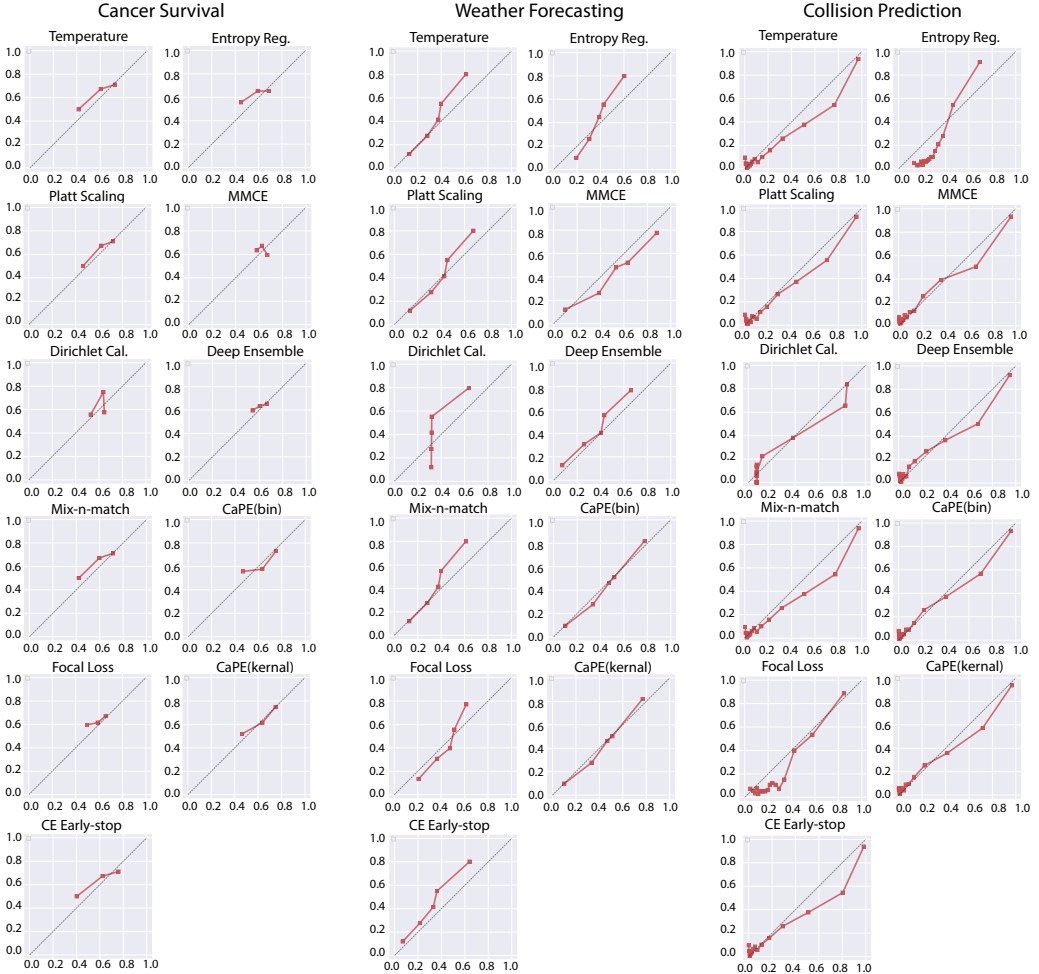

Figure 7: The reliability diagrams of all the baselines on real-world datasets. We train all baseline methods on each of the datasets and plot the empirical probability(y-axis) against predicted probability(x-axis). The axis labels are removed due to space constraints.

We present here additional reliability curves for the different synthetic data scenarios, mentioned in Figure 4.

## B   KOLMOGOROV-SMIRNOV ERROR

We derive the KS-error, mentioned in Section 3.

For a calibrated estimator

$$\mathbb{P}[y = 1 | f(\boldsymbol{x}) \in I(q)] = q, \ \ \forall 0 \leq q \leq 1,$$

for some small interval $I(q)$ around $q$.

Hence

$$\mathbb{P}[y = 1, f(\boldsymbol{x}) \in I(q)] = \mathbb{P}[f(\boldsymbol{x}) \in I(q)]q, \ \ \forall 0 \leq q \leq 1$$

. Similarly to the Kolmogorov-Smirnov (KS) test for distribution functions, we can recast this property in integral form

$$\phi_1(\sigma) = \int_0^\sigma \mathbb{P}[y = 1, f(\boldsymbol{x}) \in I(q)]dq, \quad \phi_2(\sigma) = \int_0^\sigma \mathbb{P}[f(\boldsymbol{x}) \in I(q)]qdq$$

We can evaluate $\phi_1, \phi_2$ from a finite sample $(\boldsymbol{x}_i, y_i), i = 1 \ldots n$,

$$\phi_1(\sigma) = \frac{1}{n} \sum_{i=1}^n \mathbf{1}(y_i = 1, f(\boldsymbol{x}_i) \leq \sigma), \quad \phi_2(\sigma) = \frac{1}{n} \sum_{i=1}^n \mathbf{1}(f(\boldsymbol{x}_i) \leq \sigma)f(\boldsymbol{x}_i)$$

The KS error is defined as

$$\text{KS} = \max_{1 \leq \sigma \leq 1} |\phi_1(\sigma) - \phi_2(\sigma)|$$

$\phi_1, \phi_2$ can be efficiently computed by sorting the data points with respect to their confidence scores $f(\boldsymbol{x}_i)$. The KS error has the advantage of being independent of binning configurations, unlike ECE and MCE.

## C   BRIER SCORE DECOMPOSITION

We present here a decomposition of the Brier score into two components, discussed in Section 3.

The Brier score can be interpreted as a sum of two terms, calibration and refinement. Assume the network can output one of $K$ distinct possible predictions, i.e., $\hat{p} \in \{\hat{q}_1, \ldots, \hat{q}_K\}$.

Denote $S_k$, the set of all inputs with output $p_k$ and $\bar{q}_k$ the empirical probability over $S_k$, i.e.,

$$S_k = \{\boldsymbol{x} | f(\boldsymbol{x}) = \hat{q}_k\}, \quad |S_k| = n_k, \quad \bar{q}_k = \frac{1}{n_k} \sum_{\boldsymbol{x}_i \in S_k} y_i$$

Then we can write

$$\text{Brier} = \frac{1}{N} \sum_{i=1}^N (\hat{p}_i - y_i)^2 = \frac{1}{N} \sum_{k=1}^K n_k(\hat{q}_k - \bar{q}_k)^2 + \frac{1}{N} \sum_{k=1}^K n_k \bar{q}_k(1 - \bar{q}_k),$$

The first term on the RHS, calibration, is similar to $\text{MSE}_p$, with the empirical probabilities $\bar{q}_k$ substituting for the true labels. The second term, refinement, is an estimate of the confidence in determining $\bar{q}_k$. It is related to the area under curve (AUC), which measures to the achievable accuracy of the network as a classifier. The term is smaller as the prediction classes $f_i$ tend towards 0 or 1. Thus, this term penalizes empirically calibrated predictors, with low discriminative power, as in Figure 2b.

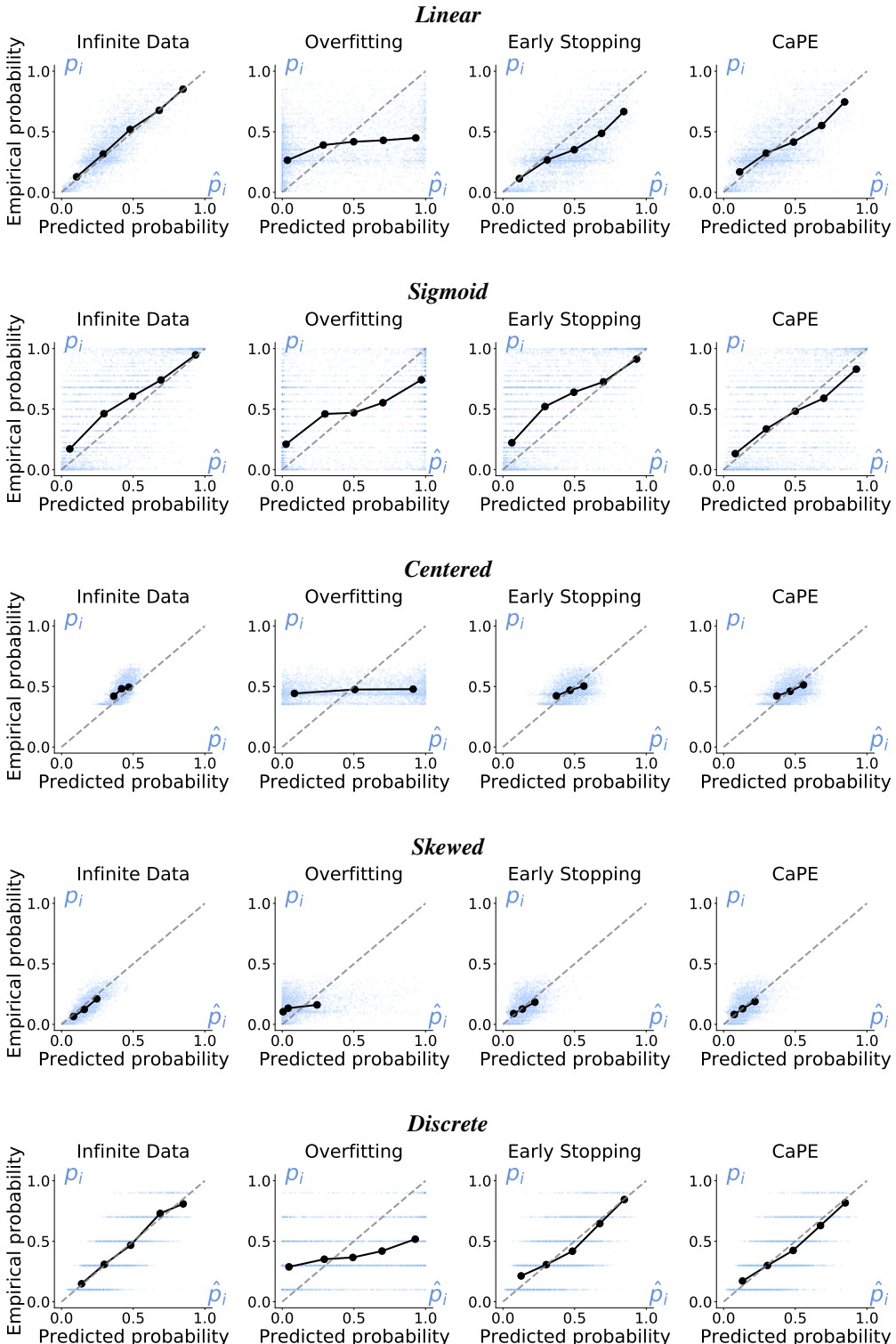

Figure 8: Reliability diagrams for different synthetic data scenarios. We can see that CaPE outperforms early stopping, prevents overfitting, and achieves a performance on par with training on infinite resampled data.

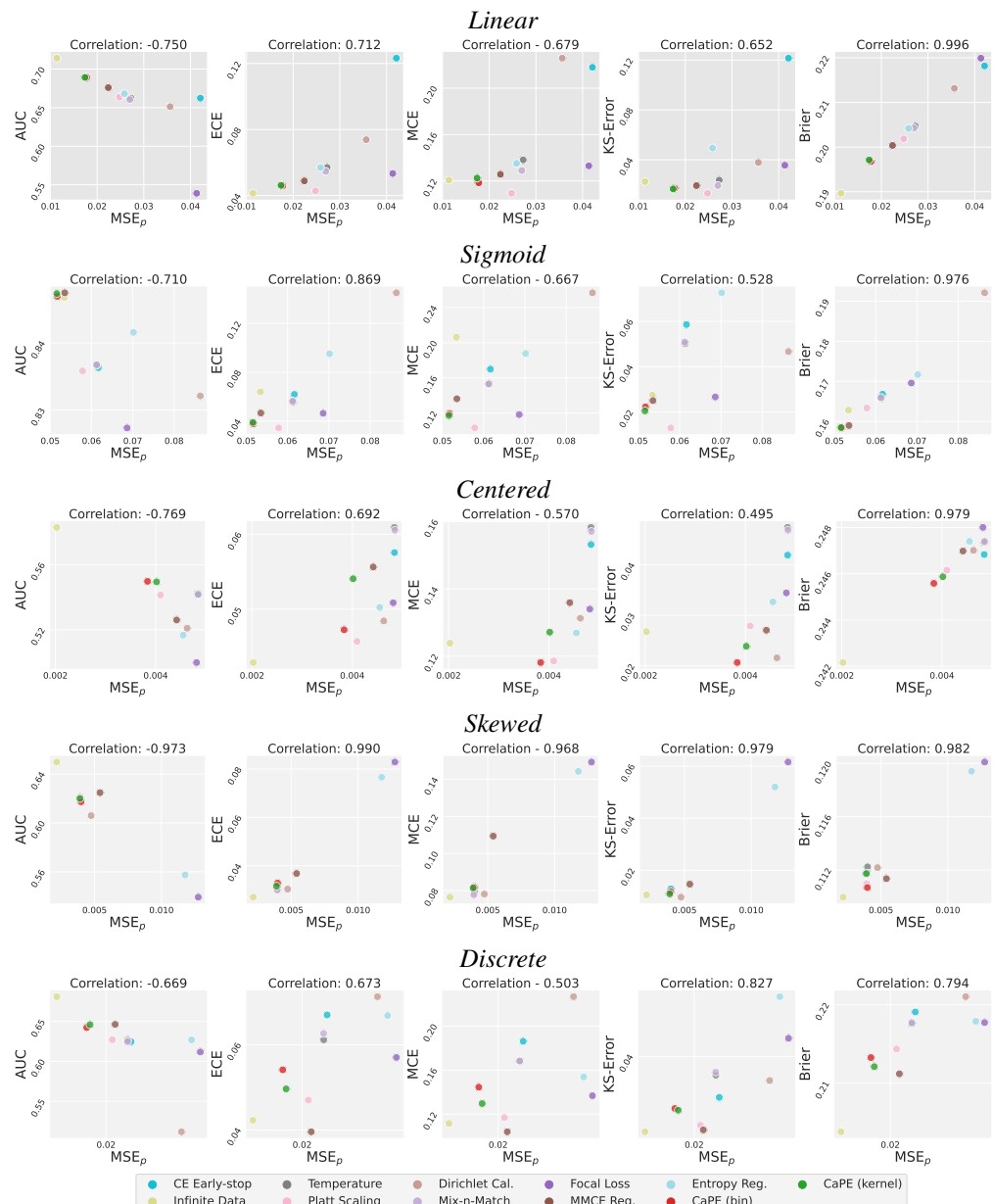

Figure 9: The comparison between $\text{MSE}_p$ and other metrics on synthetic data. Brier score presents the most consistent correlation with $\text{MSE}_p$.

## D    METRIC COMPARISON

We present here the correlation between different calibration and accuracy metrics and metrics that have access to the ground truth probabilities, $\text{MSE}_p$ and KL-divergence, eveluated over all five scenarios in our Face-based Risk Prediction synthetic dataset, referred to in Section 3.

## E    ESTIMATION OF EMPIRICAL PROBABILITY IN CaPE

We describe in further detail the two ways to estimate the conditional probability $\mathbb{P}[y = 1 | f(\boldsymbol{x}) \in I(q)]$, introduced in Section 4.

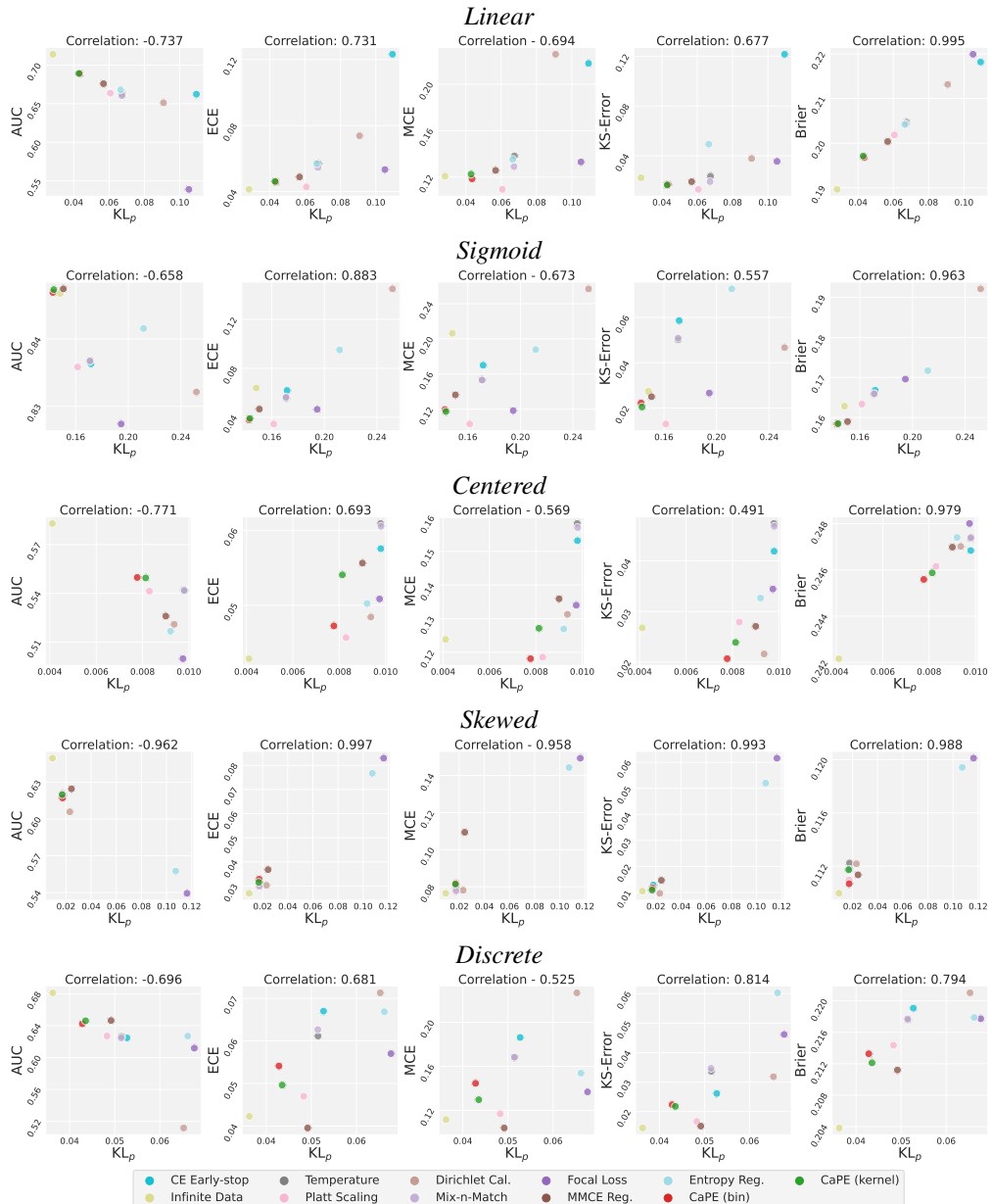

Figure 9: Comparison between $\mathrm{KL}_p$ and other metrics on synthetic data. Brier score presents the most consistent correlation with $\mathrm{KL}_p$.

We wish to estimate the conditional probability of an output $y$ given a network prediction $f(\boldsymbol{x})$, $\mathbb{P}[y = 1 | f(\boldsymbol{x}) \in I(q)]$ We can approximate the probability by averaging over points $\hat{p} \in I(q)$,

$$\mathbb{P}[y = 1 | f(\boldsymbol{x}) \in I(q)] \approx \frac{1}{|I(q)|} \sum_{\hat{p} \in I(q)} \mathbb{P}[y = 1 | f(\boldsymbol{x}) = p] \tag{7}$$

An empirical estimate of $\mathbb{P}[y = 1 | f(\boldsymbol{x}) \in I(q)]$ would be

$$\mathbb{P}[y = 1 | f(\boldsymbol{x}) \in I(q)] \approx \frac{1}{|\mathrm{Index}(I(q))|} \sum_{f(\boldsymbol{x}_i) \in I(q)} y_i, \tag{8}$$

where $\mathrm{Index}(I_q) = \{i | f(\boldsymbol{x}_i) \in I(q)\}$.

Alternatively, we can use kernel estimation:

$$\mathbb{P}[y = 1 | f(\boldsymbol{x}) \in I(q)] \approx \frac{1}{Z} \sum_{\hat{p} \in I(q)} \mathbb{P}[y = 1 | f(\boldsymbol{x}) = \hat{p}] \cdot \exp\left(-\frac{(p-q)^2}{\sigma^2}\right), \tag{9}$$

where $Z = \sum_{p \in I(q)} \exp\left(-\frac{(p-q)^2}{\sigma^2}\right)$ is the normalization factor. An empirical estimate of the conditional probability would then be

$$\mathbb{P}[y = 1 | f(\boldsymbol{x}) \in I(q)] \approx \frac{1}{Z} \sum_{f(\boldsymbol{x}_i) \in I(q)} y_i \exp\left(-\frac{(f(\boldsymbol{x}_i) - q)^2}{\sigma^2}\right). \tag{10}$$

Based on these two approximation methods, we can design an algorithm to estimate $p_{\text{emp}}^i$.

**Bin** We divide our data into $B$ bins of equal size. $Q_1, \ldots, Q_B$ are the data B-quantiles. We wish to estimate $\mathbb{P}[y = 1 | f(\boldsymbol{x}) \in [Q_{b-1}, Q_b]]$, $b = 1, \ldots, B, Q_0 = 0$. Denote $I_b := [Q_{b-1}, Q_b] \cap \{f(\boldsymbol{x}_i)\}_{i=1}^N$, set of all predictions in $[Q_{b-1}, Q_b]$, and $\text{Index}(I_b) = \{i | f(\boldsymbol{x}_i) \in I_b\}$. We have,

$$\mathbb{P}[y = 1 | f(\boldsymbol{x}) \in [Q_{b-1}, Q_b]] \approx p_{\text{emp}}^{(b)} = \frac{1}{|I_b|} \sum_{i \in \text{Index}(I_b)} y_i$$

We assign $p_{\text{emp}}^{(b)}$ to all data points $i$ in the $b$-th quantile

$$p_{\text{emp}}^i = p_{\text{emp}}^{(b)} \quad \forall i \in \text{Index}(I_b)$$

**Kernel** In this case we use kernel estimation:

$$p_{\text{emp}}^i = \frac{\sum_{k \in \text{NN}(i,r)} K(i, k) y_k}{\sum_{k \in \text{NN}(i,r)} K(i, k)}. \tag{11}$$

$\text{NN}(i, r)$ defines $r$ data points whose predictions are nearest to $\hat{p}_i = f(\boldsymbol{x}_i)$. $K(i, j)$ is the Gaussian kernel

$$K(i, j) = \exp\left(-\frac{(\hat{p}_i - \hat{p}_j)^2}{\sigma^2}\right),$$

with hyperparameter $\sigma$.

## F CALIBRATION BASELINES

This section includes a review of the baseline methods, discussed in Section 6

**Post-processing** Postprocessing for calibration requires finding a function $f : [0, 1] \rightarrow [0, 1]$, that augments the output of a the neural network $\hat{p}_i \rightarrow f(\hat{p}_i)$ in order to achieve better calibration properties

- Platt scaling (Platt, 1999) optimizes $f$ on validation set within the following family,
$$f_1(\hat{p}_i) = \sigma\left(W^T \hat{p}_i + b\right) \tag{12}$$
where $W \in \mathbb{R}^2, b \in \mathbb{R}$ and $\sigma$ is the Sigmoid function. The non-probabilistic predictions of a classifier are used as features for a logistic regression model, which is trained on the validation set to return probabilities.
- Temperature scaling (Guo et al., 2017) is a single parameter variant of Platt Scaling where we only change the temperature of the softmax to obtain the calibrated probabilities.
$$f(\hat{p}_i) = \text{Softmax}(\hat{p}_i / T) \tag{13}$$
where $T \in \mathbb{R}$ minimizes the negative log-likelihood of validation set.
- Beta/Dirchlet calibration (Dir-ODIR) (Kull et al., 2017; 2019) assumes that the probabilities can be parametrized by a Beta/Dirchlet distribution i.e.
$$f_j \sim \text{Beta}(\alpha^{(j)}, \beta^{(j)}) \tag{14}$$
Assume the prior to be $p(y = j) = \pi_j, \pi_j \in [0, 1]$, we have $P(y | f_j) \propto \pi_j f_j$, and then $\alpha^{(j)}, \beta^{(j)}$ are estimated by maximizing the posterior.

**Ensembling** These calibration methods simultaneously train several neural networks from end to end, varying parameters in the training process. The final output is some function of all the different outputs.

- Mix-n-Match (Zhang et al., 2020) improves calibration by ensembling parametric and non-parametric calibrators. Denote the temperature scaling function with $g(\hat{y}_i, T)$. Then Mix-n-Match ensembles different temperatures

$$f_j(\hat{p}_i) = w_1 g_j(\hat{p}_i, T) + w_2 g_j(\hat{p}_i, 0) + w_3 g_j(\hat{p}_i, \infty) \tag{15}$$

  After ensembling the parametric temperature scaling, Mix-n-Match applies non-parametric isotonic regression.

- Deep ensemble (Lakshminarayanan et al., 2017) trains $M$ copies of the neural network with different initialization. The probability estimation is the average of all single model estimations

$$p(y_i \mid x_i) = \frac{1}{M} \sum_j^M p_{\theta_j}(y_i \mid x_i) \tag{16}$$

**Modified training** These calibration methods train the neural networks from end to end, modifying the training process to improve calibration.

- Confidence penalty (Pereyra et al., 2017) Penalizeslow entropy output distributions (confidence penalty). Label smoothing improve state-of-the-art models across benchmarks.

$$\mathcal{L}(\theta) = -\sum_i \log p_\theta(y_i|x_i) - \beta H(p_\theta(y_i|x_i)) \tag{17}$$

- Focal loss (Mukhoti et al., 2020) maximizes entropy while minimizing the KL divergence between the predicted and the target distributions. It also regularizes the weights of the model to avoid overfitting.

$$\mathcal{L}(\theta) = -\sum_i (1 - p_\theta(y_i|x_i))^\gamma \log p_\theta(y_i|x_i), \quad \gamma \in \mathbb{R}. \tag{18}$$

- Kernel MMCE (Kumar et al., 2018) is a reproducing kernel Hilbert space (RKHS) kernel based measure of calibration that is efficiently trainable, alongside the negative likelihood loss. Given data samples $\mathcal{D} = \{(c_i, r_i)\}_{i=0}^m$, where $c_i = \chi_{\{\hat{y}_i = y_i\}}$ and $r_i = \mathbb{P}(c_i = 1|\hat{y}_i)$, MMCE is computed on samples $\mathcal{D}$ as following,

$$\text{MMCE}^2(\mathcal{D}) = \sum_{i,j} \frac{(c_i - r_i)(c_j - r_j)k(r_i, r_j)}{m^2} \tag{19}$$

  where $k(r_i, r_j)$ is a kernel function. MMCE is optimized together with the cross entropy loss as a regularization term. The strength of calibration can be adjusted by a scale $\lambda \in \mathbb{R}$.

$$\mathcal{L}(\theta) = -\sum_i \log p_\theta(y_i|x_i) + \lambda \left(\text{MMCE}^2(D)\right)^{\frac{1}{2}} \tag{20}$$

# G  SYNTHETIC DATA EXPERIMENTS

We use ResNet-18 model for all our experiments with synthetic data.

The synthetic data is split into training, validation, and test sets with 16641, 4738, and 2329 samples, respectively. The training and validation sets contain only images $x_i$ and 0-1 labels $y_i$ for training and tuning the model. In order to evaluate the performance of the model for probability estimation, the held-out test set contains the ground truth probabilities $p_i$, in addition to $x_i$ and $y_i$. Note that we do not use the ground-truth probability labels $p_i$ values during training or inference - we only use them to compare the performance of different models.

**Ground Truth Probability Generation** The ground truth probability associated with example $i$ is simulated by $p_i = \psi(z_i)$ where $z_i$ is age of the person.

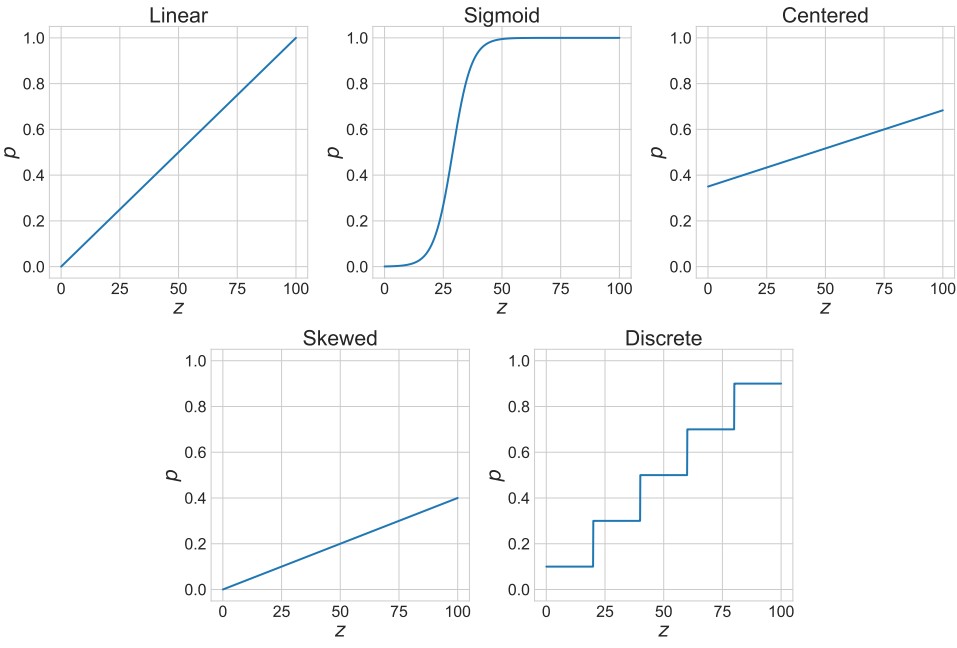

Figure 10: Illustration of the function $\psi(z)$ used to generate the different synthetic-data scenarios.

**Label distribution** After determining the probability $p_i$ using $\psi(z)$, the label $y_i$ is sampled from a Bernoulli distribution parametrized by $p_i$, so that it takes the value 1 with probability $p_i$. The distributions of $y_i$ under five different scenarios are illustrated in Fig.11.

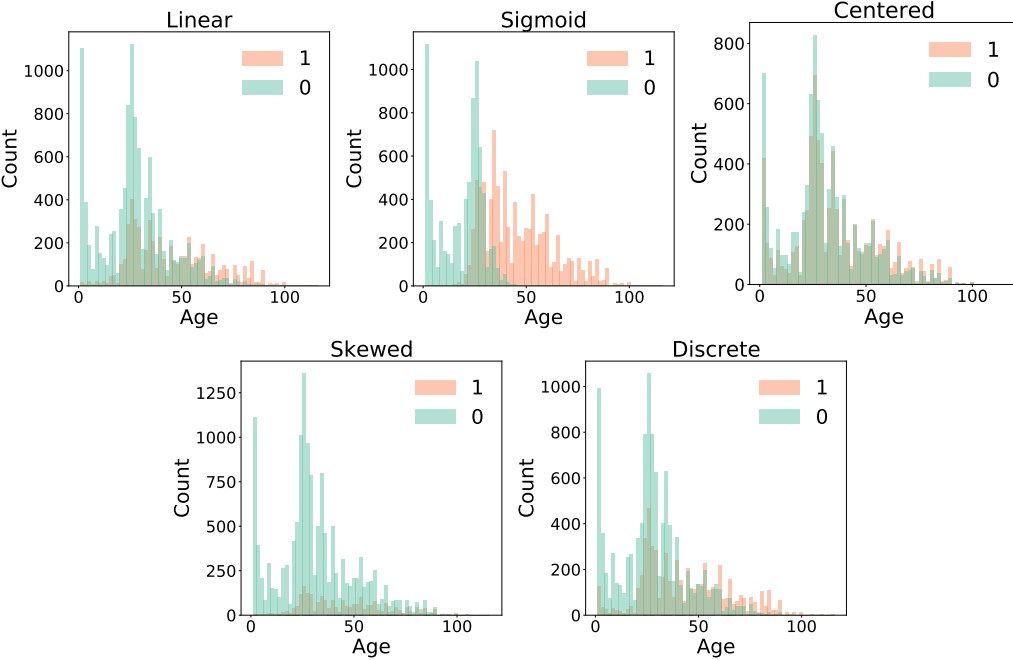

Figure 11: Histograms of the outcomes $(y_i)$ for the different synthetic-data scenarios.

## H   REAL-WORLD DATA AND EXPERIMENT DETAILS

We present here supplementary information for the real-world datasets used in our experiments.

**Cancer Survival**   Histopathological features are useful in identification of tumor cells, cancer sub-types, and the stage and level of differentiation of the cancer. Hematoxylin and Eosin (H&E)-stained slides are the most common type of histopathology data and the basis for decision making in the clinics. With these properties, H&E are used for mortality prediction of cancer (Wulczyn et al., 2020). In this experiment, we use the H&E slides of non-small cell lung cancers from The Cancer Genome Atlas Program (TCGA)[4] to predict the 5-year survival. The dataset has 1512 whole slide images from 1009 patients, and 352 of them died in 5-years. We split the samples by patients and source institutions into training, validation, and test set, which has 1203, 151, and 158 samples respectively.

The whole slide images contain numerous pixels, so we cropped the slides into tiles at 20x magnification with 1/4 overlapping, resized them to $299 \times 299$ with bicubic interpolation, and filtered out the tiles with more than 85% area covered by the background. The representations of each tile are trained with self-supervised momentum contrastive learning (MoCo) (Chen et al., 2020), and the slide-level prediction is obtained from a multiple-instance learning network (Ilse et al., 2018) trained with the binary label of survival in 5 years.

**Weather Forecasting**   We use the German Weather service dataset[5], which contains quality-controlled rainfall-depth composites from 17 operational Doppler radars. Three precipitation maps from the past 30 minutes serve as an input. The training labels are the 0/1 events indicating whether the mean precipitation increases (1) or not (0).

The German Weather service (DWD - Deutshce Wetter Dienst) dataset `https://opendata.dwd.de/weather/radar/` contains quality-controlled rainfall-depth composites from 17 operational DWD Doppler radars. It has a spatial extent of 900x900 km, and covers the entirety of Germany. Data exists since 2006, with a spatial and temporal resolution of 1x1 km and 5 minutes, respectively. The dataset has been used to train RainNet, a pricipitation nowcasting model (Ayzel, 2020).

The network architecture is ResNet18, with 3 input channels and 2 output channels. The input to the network are 3 precipitation maps which cover a fixed area of 300km×300 km in the center of the grid (300× 300 pixels), set 10 minutes apart. The training, validation and test datasets consist of 20000, 6000 and 3000 samples, respectively, all separated temporally, over the span of 15 years.

**Collision Prediction**   Vehicle collision is one of the leading causes of death in the world. Reliable collision prediction systems which can warn drivers about potential collisions can save a significant number of lives. A standard way to design such a system is to train a convolutional model for identifying if a particular vehicle in the dash-cam video feed might collide with the car in next few seconds. More formally, at time $t = T$ the system tries to predict if any car in the video might collide with our given car in time $t \in [T, T + T_{\text{look-ahead}}]$. Each labelled training sample consists of features $X = (X^{T-\delta}, X^{T-2\delta}, \ldots, X^{T-d\delta})$ and a binary label $Y \in \{0, 1\}$ denoting if an accident will occur in $t \in [T, T + T_{\text{look-ahead}}]$. Each $X^t$ is a tensor with 4 channels where the first 3 channels corresponds to an RGB image of the dashcam view at time $t = t$, and the fourth channel consists of a mask with a bounding box on a particular vehicle of interest. In this work, we use *YouTubeCrash* dataset (Kim et al., 2019) to train and test our model, which uses $\delta = 0.1s$, $T_{\text{look-head}} = 18\delta = 1.8s$, and $d = 3$. Following Kim et al. (2019) we used a VGG-16 network architecture.

The dataset contains 122 accident scenes, and 2096 non-accident scenes, which after feature extraction gives us 2096 positive samples, and 11486 negative samples (the dataset is severely imbalanced, and similar to the Skewed situation in Section 6.1). We further split the dataset into train (6453 samples for label 0, and 1023 samples for label 1), validation (2348 samples for label 0, and 545 samples for label 1), and test (2685 samples for label 0, and 528 samples for label 1) sets. The samples in train, validation and test sets are generated from disjoint scenes/dashcam videos.

---

[4]`https://www.cancer.gov/tcga`
[5]`https://opendata.dwd.de/weather/radar/`

# I ANALYSIS OF CANCER SURVIVAL RESULTS

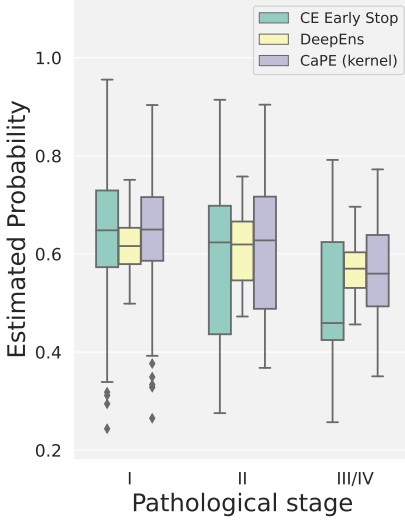

Figure 12: Estimated probability of survival grouped by pathological stages. The plot shows median, samples between 25th to 75th percentile in the box, samples between 0th and 100th percentile on the line, and the outliers as dots. Deep ensemble produces similar probability estimates for patients across all the stages; CE is more discriminative but has a very large variance; CaPE achieves a trade-off between the two baselines.

For cancer survival prediction, we visualize the estimated probabilities on the test set in different pathological stages in Figure 12. In general, patients in earlier stages should have higher probabilities of survival. Deep ensemble produces similar probability estimates for all stages (i.e the model is less discriminative). Cross-entropy minimization (CE) is more discriminative, but has very wide confidence intervals. CaPE is more discriminative than deep ensemble, while having narrower confidence intervals than CE.

# J CALIBRATING FROM THE BEGINNING

CaPE exploits a calibration-based cost function to improve its probability estimates without overfitting. The empirical probabilities in this loss are computed from the model itself. Consequently, applying this strategy from the beginning of training can be counterproductive, because the model predictions are essentially random. This is demonstrated in the following table, which compares CaPE with a model trained using the calibration loss from the beginning (in the same way as CaPE, alternating with cross-entropy minimization).

| Methods | Linear | | Sigmoid | | Centered | | Skewed | | Discrete | |
|---|---|---|---|---|---|---|---|---|---|---|
| ($\times 10^{-2}$) | $MSE_p$ | $KL_p$ | $MSE_p$ | $KL_p$ | $MSE_p$ | $KL_p$ | $MSE_p$ | $KL_p$ | $MSE_p$ | $KL_p$ |
| Bin (start) | 2.59 | 6.81 | 8.07 | 22.10 | 0.48 | 0.98 | 0.51 | 2.37 | 2.74 | 6.36 |
| Kernel (start) | 2.23 | 5.68 | 7.60 | 21.15 | 0.54 | 1.10 | 0.68 | 2.84 | 2.40 | 5.63 |
| Bin (CaPE) | 1.83 | 4.46 | 5.29 | 14.59 | **0.38** | **0.78** | 0.40 | 1.72 | **1.83** | **4.31** |
| Kernel (CaPE) | **1.81** | **4.41** | **5.22** | **14.47** | 0.40 | 0.81 | **0.39** | **1.70** | 1.85 | 4.36 |

Table 11: Comparison between CaPE and a model that uses the calibration loss from the beginning (in the same way as CaPE, alternating with cross-entropy minimization) on synthetic data.

## K COMPARISON OF REAL-WORLD DATASETS WITH DIFFERENT SCENARIOS OF THE SIMULATED DATASET

Figure 13 illustrates the similarity between the empirical probability curves of different real-world datasets and the different scenarios of our synthetic dataset. For the cancer survival dataset, the empirical probabilities are clustered in the center (0.4-0.6) similar to the *Centered* scenario. For the weather forecasting dataset, the probabilities are uniformly distributed across 0.1-0.8 similar to the *Linear* scenario. For the collision prediction dataset, the majority of the data points are clustered in the lower probability region which makes it similar to the *Skewed* scenario.

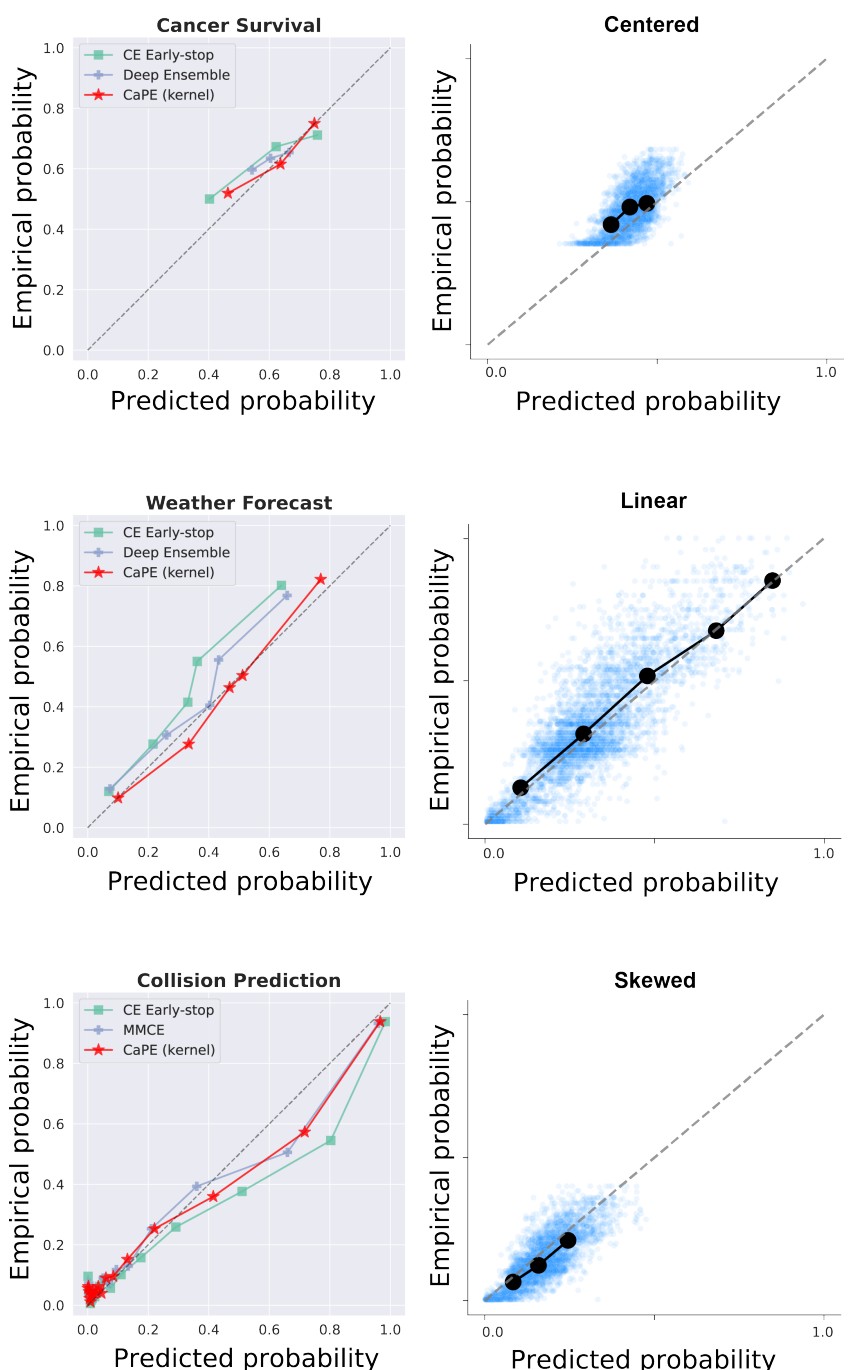

Figure 13: **Comparison of reliability diagrams for real-world data with different scenarios of simulated data**. For the cancer survival dataset, the empirical probabilities are clustered in around (0.4-0.6), similar to the *Centered* scenario. For the weather forecasting dataset, the probabilities are uniformly distributed across 0.1-0.8, similar to the *Linear* scenario. For the collision prediction dataset, the majority of the output probabilities are clustered in the lower probability region, similar to the *Skewed* scenario.

