# OpenReview forum: "Deep Probability Estimation"
_ICLR.cc/2022/Conference — ICLR 2022 Submitted_

### Official Review · Reviewer_Lp12 · 2021-11-01

**Correctness:** 3
**Technical Novelty And Significance:** 3
**Empirical Novelty And Significance:** 3
**Recommendation:** 5
**Confidence:** 4

**Main Review:**

The authors address an interesting problem. However, it would be interesting to further justify why current models do not offer an accurate estimate of uncertainty, particularly bayesian modelling. It is also important to note that calibration is a key issue in the survival literature and penalties have been developed to tackle this issue [1].

An interesting point is the decomposition of the Brier loss, which would benefit from further descriptions.

Finally, it is interesting that the authors use their loss only as a fine-tuning of the neural network. From figure 5, it seems that the introduced loss does not allow to reach a lower minimum but stops the training of the neural network. It would be interesting to see what is the effect of training the neural network from the start with this given loss.

[1] Lee, C., Zame, W.R., Yoon, J. and van der Schaar, M., 2018, April. Deephit: A deep learning approach to survival analysis with competing risks. In Thirty-second AAAI conference on artificial intelligence.

**Summary Of The Paper:**

This work tackles the problem of probability estimation. Current machine learning models do not fully reflect the uncertainty of the outcome but only of the model. The authors propose a loss that enforces both calibration and discrimination. Additionally, they present a semi-synthetic dataset for further study of this problem: from an image dataset of faces with associated age, they estimate the risk of developing a given disease.

**Summary Of The Review:**

The work is clear and concise. The authors propose to tackle an interesting problem. However, it would benefit from further justifications of the utility and contextualisation with other domains of machine learning.

---

> ### Author Response · Authors · 2021-11-19
> **Response to reviewer Lp12**
>
> We thank the reviewer for their thoughtful and valuable comments and we appreciate the reviewer for pointing out that we are addressing an interesting problem. We address each specific comment in detail below.
>
> **1. Justify why current models do not offer an accurate estimate of uncertainty.**
> In the paper, we show that several methods developed for calibration of classification models do in fact produce reasonably accurate uncertainty estimates for probability estimation problems, although they are consistently outperformed by the proposed method.
>
> A method that does not offer an accurate estimate of uncertainty is training a cross-entropy loss, without performing early stopping. When the network is trained for long enough, the model eventually “memorizes” the 0-1 labels corresponding to the outcomes. As illustrated in the second column of Figure 4, the predicted probability eventually collapses to 0 and 1, in such a way, the **training** loss goes to zero (See the blue curve in the first column of Figure 5). This is not sensitive to model architectures and is indeed a property of minimizing cross-entropy loss on 0-1 labels with high-capacity deep neural networks.
>
> We thank the reviewer for the suggestion of referring to Bayesian modeling, we have added some references. We note that we have included a related method (deep ensemble) in our baseline methods.
>
> **2. Calibration is a key issue in the survival analysis literature.**
> We agree with the reviewer that the connection with the survival analysis problem is interesting and now discuss this in the conclusion.
>
> **3. Decomposition of the Brier loss.**
> We thank the reviewer for raising this interesting point. We have included a brief discussion in Appendix C.
>
> **4. The proposed loss only finetunes the neural network.**
> It is true that the proposed method “does not allow to reach a lower minimum” on **training data**. However, this is actually desired as continuing minimizing the cross-entropy loss on 0/1 labels would result in collapsed probability estimates, and hurt generalization on the validation set. As shown in Figure 5, when the training loss keeps dropping, the validation loss instead increases, suggesting overfitting and collapsed probability estimates. In contrast, the proposed loss is able to further decrease the validation loss, preventing overfitting and improving the probability estimates.
>
> **5. It would be interesting to see what is the effect of training the neural network from the start.**
> We agree that this is interesting. Our intuition is that at the beginning, the probability estimates produced by the model are essentially random. As a result, enforcing calibration based on these probabilities would not be beneficial. The experiment results (see the table below) consistently show lower performances if the proposed loss is applied from the start of training (first two rows), in line with our intuition. We believe that this is a nice motivation for our proposed strategy of first training solely using cross-entropy, and then alternating with minimization of the calibration loss, once the probability estimates are meaningful. We have included the results in the appendix (Section J).
>
> |                    | Linear || Sigmoid || Centered || Skewed || Step ||
> | ------------------ | :------: | :------: | :-------:| :-------: | -------- | -------- | ------ | ------ | ------ | ----- |
> | Method             | MSE\_p | KL\_p  | MSE\_p  | KL\_p   | MSE\_p   | KL\_p    | MSE\_p | KL\_p  | MSE\_p | KL\_p |
> | Bin (start)        | 2.59   | 6.81   | 8.07    | 22.1    | 0.48     | 0.98     | 0.51   | 2.37   | 2.74   | 6.36  |
> | Kernel (start)     | 2.23   | 5.68   | 7.6     | 21.15   | 0.54     | 1.1      | 0.68   | 2.84   | 2.4    | 5.63  |
> | Bin (fine-tune)    | 1.83   | 4.46   | 5.29    | 14.59   | **0.38**     | **0.78**     | 0.4    | 1.72   | **1.83**   | **4.31**  |
> | Kernel (fine-tune) | **1.81**   | **4.41**   | **5.22**    | **14.47**   | 0.4      | 0.81     | **0.39**   | **1.7**    | 1.85   | 4.36  |
>
> **6. Utility and contextualization with other domains of machine learning**
> We thank the reviewer for the suggestion. We have edited the paper to highlight the utility of probability estimation in several applications (connected to the real-world datasets that we use). In addition, we have elaborated on connections to calibration in classification problems (in the introduction), classification from noisy labels (in the second paragraph of Section 4), and survival analysis (in the conclusion).

---

> > ### Comment · Reviewer_Lp12 · 2021-11-29
> > **Response**
> >
> > Thank you for the clarifications
> >
> > However, after consideration of the other reviewers’ comments, I am leaning towards weak rejection as the paper would gain from further clarification of the problem definition.

---

> > > ### Author Response · Authors · 2021-11-29
> > > **Response**
> > >
> > > Thank you for your feedback. In the revised paper we have addressed other reviewers’ comments to clarify the problem statement, which is now discussed in further detail in the abstract, introduction, and problem statement sections of the paper. Could you please let us know what further clarification is necessary or what concrete details are missing?
> > >
> > > The problem definition is currently explained:
> > >
> > > (1) In the abstract:
> > > “Probability-estimation models are trained on observed outcomes (e.g. whether it has rained or not, or whether a patient has died or not), because the ground-truth probabilities of the events of interest are typically unknown. The problem is therefore analogous to binary classification, with the important difference that the objective is to estimate probabilities rather than predicting the specific outcome. The goal of this work is to investigate probability estimation from high-dimensional data using deep neural networks.”
> > >
> > > (2) In the introduction: paragraph 1 provides concrete examples tied to the datasets we use.
> > >
> > > (3) In Section 2, which is completely devoted to explaining the problem formulation.
> > >
> > > (4) In Figure 1, which illustrates the problem formulation using a concrete example.
> > >
> > > (5) In Figure 2, which illustrates that good calibration is not enough for accurate probability estimation.

---

### Official Review · Reviewer_vz2e · 2021-11-02

**Correctness:** 4
**Technical Novelty And Significance:** 2
**Empirical Novelty And Significance:** 2
**Recommendation:** 5
**Confidence:** 4

**Main Review:**

The definition of of ECE and MCE is a special case (for binary classification) of the more general one that handles multi-class classification. Mathematically the definition is accurate, but overly obscure and complicated. I suggest the authors refer to the definition from Guo et al., 2017. Essentially, ECE calculate the difference between confidence and accuracy in each bin. It is therefore a bit confusing to call the accuracy term p_emp.

In term of methodology, it makes sense to alternate between the two losses. However, it seems CaPE uses the training set to compute p_emp. Would p_emp itself run into the problem of over-confidence? For example, a well trained neural network (NN) is always over-confident and therefore most predictions (probability) will concentrate around 1, meaning that most bins will have nearly zero data point. Could you provide more intuition or theoretical analysis on why over-confidence would not happen?

Why is the discrimination loss only one-sided? In other words, what happens if the normal cross-entropy loss is used?

This brings me to another concern which is the extension to multi-class classification. The current discrimination loss will not work probably when there are more classes since it only focuses on the single correct class. Therefore, I would expect that the performance will get worse and worse as the number of classes increases.

The plots in Figure 5 show the benefit of CaPE. However, the drop of training loss and increase of validation loss are quite steep for CE. In practice, this rarely happens for high dimensional data, especially if the network architecture is well chosen.

The baselines seem comprehensive. Are there results for ECE/MCE/Brier Score for Table 1?

In Table 2, it seems CaPE is worse than Deep Ensemble and MMCE Reg. in Collision Prediction and worse than Deep Ensemble  in Weather Forecasting (in terms of AUC). I was wondering how many models are used for Deep Ensemble, since the performance of Deep Ensemble is also related to the number of individual models.



Minor:
There are two entries for the same reference Gupta et al., 2021.

Some related work on probabilistic neural network for uncertainty estimation is missing [a,b,c].

[a] Natural-Parameter Networks: A Class of Probabilistic Neural Networks, NIPS 2016
[b] Feed-forward Propagation in Probabilistic Neural Networks with Categorical and Max Layers, ICLR 2018
[c] Sampling-free Epistemic Uncertainty Estimation Using Approximated Variance Propagation, CVPR 2019


**Summary Of The Paper:**

The paper tackles the problem of uncertainty calibration in the special case of binary classification. The authors propose the so-called CaPE to iteratively minimize the discrimination loss and the calibration loss.

**Summary Of The Review:**

Overall, the authors proposed a simple and effective method for improving the uncertainty estimation for binary classification. Several drawbacks include: the reason why training directly on the training will not lead to over-confidence is unclear; performance is worse than some baselines in some dataset; the method seems limited to binary classification and it would be interesting to see at least its performance on multi-class classification.

---

> ### Author Response · Authors · 2021-11-19
> **Response to reviewer vz2e (1/2)**
>
> We thank the reviewer for the detailed comments. We address each of them below.
>
> **1 - Mathematically the definition (ECE and MCE) is accurate but overly obscure and complicated. I suggest the authors refer to the definition from Guo et al., 2017. Essentially, ECE calculates the difference between confidence and accuracy in each bin. It is therefore a bit confusing to call the accuracy term p_emp.**
>
> We appreciate the reviewer pointing out another definition of calibration error (difference between accuracy and confidence) in Guo et al. 2017. We would like to clarify the difference between the two definitions and why Definition 3.1 is more appropriate in this paper.
> Guo et al. focus on classification problems and assume that the model outputs (1) predictions of the outcome (in our case, 0/1) and (2) confidence, which is a probability quantifying how confident the model is of this prediction. Mathematically, as pointed out by the reviewer, we can establish an equivalence between their definition and ours if we set our “predictions” to all equal 1, and our confidence to equal the probability output by the model. In this case, accuracy corresponds to our definition of empirical probability. However, if the predictions are not set to 1, then the definitions are not the same. If we have two examples that result in probability estimates equal to 0.1 and 0.9, then their class predictions should be 0 and 1, and confidence scores should be 0.9 and 0.9, respectively. Binning them together based on confidence does not yield an empirical probability that is appropriate for probability estimation. Instead, we should bin the estimated probabilities following Definition 3.1, which does yield an appropriate empirical estimate. We thank the reviewer for bringing this up, we have added a small discussion in the paper to prevent confusion.
>
> Another reason for introducing definition 3.1 is that it provides the general form of calibration error. ECE can be viewed as the L_1 form of this error, MCE can be viewed as the L_\infinity form of the error. The calibration loss used in CaPE also comes from Definition 3.1, which can be viewed as the cross-entropy form of the error. It provides a mathematical intuition for why we incorporate the calibration loss in CaPE to improve probability estimation.
>
>
> **2 - However, it seems CaPE uses the training set to compute p_emp. Would p_emp itself run into the problem of over-confidence? Could you provide more intuition or theoretical analysis on why over-confidence would not happen?**
>
> We completely agree with the reviewer that this can happen. In probability estimation, we observe that neural networks indeed eventually overfit the observed outcomes completely. Moreover, the estimated probabilities collapse to 0 or 1 (Figure 4, second column), a phenomenon that has also been reported in classification (Mukhoti et al., 2020). Through our numerical experiments, we show that this phenomenon is mitigated via early stopping, which yields relatively good calibration (see for example Figures 4 and 8, third column)”. To be clear, by early stopping, we mean selecting the model with respect to the best validation loss, which avoids the predicted probability collapsing to 0 and 1.
>
> Intuitively, the reason why the model does not become overconfident at first before early-stopping, is that it is simpler to learn a model that produces reasonable probabilities, but given the high capacity of deep networks, eventually, the model is capable of overfitting the outcomes. This is reminiscent of training deep neural networks on data with noisy labels (i.e. incorrect annotations). When trained on noisy labels, deep neural networks are known to eventually overfit the examples with false labels [1]. However, before that, the training data with clean labels are fitted during an early learning phase. In this work, we show that a similar early learning phase occurs in probability estimation as well.
>
> CaPE avoids over-confidence by fine-tuning the network based on the early-stopped checkpoint with the lowest validation loss. As a result, the probabilities estimated by the model are somewhat accurate, and they can be incorporated in the calibration loss which prevents the model output from becoming overconfident (see the training curves in Figure 5).
>
>
> **3 - that most bins will have nearly zero data point**
>
> Our bins each contain an equal number of examples. They are not chosen to have equal width but rather correspond to quantiles of the output probabilities.

---

> > ### Author Response · Authors · 2021-11-19
> > **Response to reviewer vz2e (2/2)**
> >
> > **4 - Why is the discrimination loss only one-sided? In other words, what happens if the normal cross-entropy loss is used?**
> >
> > This was a typo. Thanks for the careful reading! We actually used normal cross-entropy as our discriminative loss as stated in text “Cross entropy between the model output and the observed binary outcomes”. We have corrected the equation to L_C = −\sum_{i=1}^n y_i log(f(x_i)) + (1-y_i) log(1-f(x_i))], which makes it consistent with the text.
> >
> >
> > **5 - the drop of training loss and increase of validation loss are quite steep for CE**
> >
> > We tuned different learning rates and schedulers when training the CE model. Some other options (like smaller learning rates with no scheduler) did result in flatter learning curves, but their lowest validation loss was greater than the model reported. In the paper, we presented the plots of our best CE model.
> >
> >
> > **6 - The baselines seem comprehensive. Are there results for ECE/MCE/Brier Score for Table 1?**
> >
> > For the synthetic experiments, we have access to the ground-truth probabilities, so we mainly compare the differences between our model predictions and ground truth probabilities using MSE_p and KL_p. These are our “gold-standard” metrics for probability estimation, but we also included the other metrics in Appendix A.
> >
> >
> > **7 - I was wondering how many models are used for Deep Ensemble**
> >
> > We use 5 as the size of the ensemble, as in most of the experiments in [2,3]. The ablation studies in [2,3] also show that the improvement after 5 networks is marginal, compared to 1-4 networks.
> >
> >
> > **8 - it seems CaPE is worse than Deep Ensemble and MMCE Reg. in Collision Prediction and worse than Deep Ensemble in Weather Forecasting (in terms of AUC).**
> >
> > The only method that outperforms CaPE in AUC over all our experiments is deep ensemble for weather forecasting (79.86 vs 78.99). Deep Ensemble and MMCE achieve slightly better ECE metric in collision prediction. However, for these real-world datasets,  we have found that the metric that best quantifies the quality of probability estimates is the Brier score. Our experiments on the simulated dataset, where we have access to the ground-truth probabilities, allow us to establish that out of all the metrics (ECE, MCE, KS-error, AUC, Brier score) the Brier score is the one that most closely correlates with the gold-standard metrics like MSE_p and KL_p (see Fig. 3 and Appendix D) in varied scenarios. CaPE outperforms the other baseline methods on the Brier score in all the real-world datasets and all the synthetic-data scenarios.
> >
> >
> > **9 - reference**
> >
> > We corrected the duplicated citation on Gupta et al. 2021. We also thank the reviewer for their suggestions and have included them.
> >
> >
> > **10 - the method seems limited to binary classification and it would be interesting to see at least its performance on multi-class classification.**
> >
> > We would like to emphasize that we are not focused on classification problems, but rather in probability estimation. However, we agree that a generalization to multiclass situations would be interesting. This would be appropriate for problems with multiple possible outcomes with different probabilities (for example, in cancer survival this could be “survived without recurrence”, “survived with recurrence”, or “death”). In all our datasets the outcomes are binary, so this multiclass extension is not appropriate. We now mention this in the discussion.
> >
> >
> >
> > Reference
> >
> > [1] Liu, Sheng, Jonathan Niles-Weed, Narges Razavian, and Carlos Fernandez-Granda. "Early-Learning Regularization Prevents Memorization of Noisy Labels." Advances in Neural Information Processing Systems 33 (2020).
> >
> > [2] Lakshminarayanan, B., Pritzel, A., & Blundell, C. (2017). Simple and scalable predictive uncertainty estimation using deep ensembles. Advances in neural information processing systems, 30.
> >
> > [3] Fort, Stanislav, Huiyi Hu, and Balaji Lakshminarayanan. "Deep ensembles: A loss landscape perspective." arXiv preprint arXiv:1912.02757 (2019).

---

> ### Comment · Reviewer_vz2e · 2021-11-29
> **Thanks for the response**
>
> I have read the authors’ response as well as other reviewers’ comments. I appreciate the authors’ detailed feedback. However, my concern on the difference and connection between the so called ‘probability estimation’ and typical ‘uncertainty estimation/quantification’ is not well addressed. I agree with Review r2Pt on related comments.
>
> In terms of the difference between typical calibration error and the proposed metric, the author mentioned that ‘If we have two examples that result in probability estimates equal to 0.1 and 0.9, then their class predictions should be 0 and 1, and confidence scores should be 0.9 and 0.9, respectively. Binning them together based on confidence does not yield an empirical probability that is appropriate for probability estimation.’ This is not true. In typical calibration error, one does not treat them as 0.1 and 0.9 and bin them separately; instead, one treats it as a multi-class setting (in this case only 2 classes) and therefore has the confidence score of 0.9 and 0.9, which is exactly the same as the proposed ‘probability estimation’.
>
> I therefore would like to keep my score unchanged. However, I do encourage the author to dig deeper and possibly into the multi-class setting where the connection and difference would be more apparent.

---

> > ### Author Response · Authors · 2021-11-29
> > **Clarification**
> >
> > Thank you for your feedback. We would like to clarify the example you mentioned in the feedback. We wrote "**probability estimates** equal to 0.1 and 0.9, then their class predictions should be 0 and 1, and **confidence scores** should be 0.9 and 0.9, respectively". It exactly matches what you wrote **"therefore has the confidence score of 0.9 and 0.9"**.
> >
> > We are confused why it's not true.

---

> > ### Comment · Reviewer_vz2e · 2021-11-29
> > **Follow-up**
> >
> > Thanks for the clarification. This is helpful. I guess in this case, the difference between the so-called 'probability estimation' and 'uncertainty calibration' is that the former is **asymmetric** in terms of different classes. This begs another question of why this is needed (motivation) and the exponentially growing dimensions when considering multi-class problems (scalability).

---

### Official Review · Reviewer_r2Pt · 2021-11-04

**Correctness:** 1
**Technical Novelty And Significance:** 1
**Empirical Novelty And Significance:** 1
**Recommendation:** 1
**Confidence:** 4

**Main Review:**

The main message of the paper is not clear. What is essentially the mathematical definition of probability estimation? All definitions in Eqs 1 - 6 are well-established terms in the uncertainty calibration literature and the phenomenon in Fig 2 is also well known to the community. It is accounted for simply as reporting prediction accuracy and calibration scores together. I am then missing what is new thing the "probability estimation" notion is telling us and where we see it in the experiments, which are
designed simply as standard calibration experiments.

I am having hard time to make sense of the calibration loss presented as an alternative to the discrimination loss. Where does p_emp^i come from? How can one define it without giving reference to a sample-based estimation of the confidence score of f(x), which eventually boils down to the calibration we know? Having given a reference to a confidence score estimate, how is it different then from ordinary calibration?

The "function kernel" inside Algorithm 1 appears to be the only part where one can argue about a methodological novelty. This function only replaces the Dirac measure used by standard binning with a kernel density estimator. This is a standard trick to improve numerical stability, but it is hard to say that this much makes a scientific novelty worthwhile to be published as a main-track conference paper.

The paper lacks a clear focus in its core theme and a consistent logical flow. Is the main message how good Algorithm 1 is? Then the justification of the algorithm and explanation of why it works in the way it works is missing. Is it the allegedly new applications where uncertainty calibration is used? Then references to the closest work are missing. I also do not think these are such novel applications of uncertainty calibration. For instance see:

Thagaard et al., Can you trust predictive uncertainty under real dataset shifts in digital pathology?, MICCAI 2020

**Summary Of The Paper:**

The paper studies the problem of uncertainty quantification in deep neural nets. It introduces a concept called "probability estimation" and an uncertainty calibration method called "CaPE" based on this new concept. The studies uncertainty calibration in a few new data sets such a histopathology cancer diagnostics.

**Summary Of The Review:**

The paper lacks a clear problem statement, sufficient technical novelty, and a consistent logical flow. Under these conditions, I am not able to recommend an accept.

---

> ### Author Response · Authors · 2021-11-19
> **Response to reviewer r2Pt (1/3)**
>
>
> Thank you for the comments. We feel that we have not made sufficiently clear (1) the distinction between calibration and probability estimation, and (2) our methodological contributions. Our focus is on the problem of probability estimation, which can look misleadingly similar to binary classification, but is inherently different, as explained in detail below. The current literature on calibration focuses on evaluating uncertainty stemming from model accuracy, rather than inherent uncertainty associated with the problem data. In the paper we show that methods designed for calibration do indeed work to some extent for probability estimation (as correctly suggested by the reviewer). To the best of our knowledge, previous works have not shown this for tasks with inherent uncertainty. Additionally, we propose a novel method that outperforms these methods across the different datasets. We appreciate the reviewer’s comments, and have made modifications to the paper to address them. We would be very grateful if they could reconsider their evaluation taking this into account.
>
> Please see detailed responses to specific comments below:
>
> **What is essentially the mathematical definition of probability estimation? All definitions in Eqs 1 - 6 are well-established terms in the uncertainty calibration….  I am then missing what is new thing the "probability estimation" notion is telling us and where we see it in the experiments, which are designed simply as standard calibration experiments.**
>
> Thank you for bringing this to our attention. We agree that there could be confusion between probability estimation and calibration in classification. In probability estimation we are interested in working on problems that have “inherent” uncertainty, which is different from calibration uncertainty in classification. We have edited the paper to make this distinction clearer.
>
> $\textbf{Calibration uncertainty in classification}$:
>
> In typical classification applications such as image recognition in benchmarks datasets such as MNIST, CIFAR-100, or ImageNet, the label associated with each example is determined according to the content of the image. An image that contains a cat is labelled as “cat”. There is no uncertainty about whether there is a cat or not. The image label (cat/airplane/dog etc.) is deterministic and the label assignment is unambiguous. The goal of uncertainty estimation for this kind of dataset is to quantify uncertainty stemming from limited model accuracy, i.e. if we take all images such that the model claims have 80% probability of containing a cat, do 80% of them indeed contain a cat. This 80% does not reflect actual uncertainty in whether there is a cat or not in the data (either there is or there isn’t), but rather uncertainty about the model prediction.
>
> $\textbf{Probability estimation}$:
>
> In this paper, we consider problems where the outcome itself is inherently uncertain. Hence, the source of uncertainty is not the model alone. In contrast to classification, the label indicates an outcome that is not fully determined by the input data. The three real-world applications we focus on in the paper provide specific examples. Given a certain histopathology image a cancer patient may or may not die in the next five years. Given a certain radar image it may or may not rain 30 minutes later. Given a certain dashboard image, there may or there may not be an accident a few seconds after. The specific outcome has inherent uncertainty because it depends on factors that are not captured in the data (for example, comorbidities or other health factors in cancer survival prediction, wind changes in weather prediction, and road conditions or driver expertise in collision prediction). However, the data does capture important information that makes it possible to determine the probability of the outcome given the observed data.
>
> In more detail, consider the cancer-survival application:  In that case, each input image $x_i$ is a histopathology image of a patient, and $y_i$ equals 1 if the patient survived for five years after $x_i$ was collected.  The patient's survival y_i does not depend deterministically on the input data $x_i$, $\textbf{instead the probability that}$ $ y_i = 1$ $\textbf{depends on the input data}$ $x_i$. Consequently, our dataset contains very similar images that have different associated outcomes. The problem can therefore not be interpreted as a classification problem, because the labels are not completely predictable. Instead, the goal is to predict the probability of the outcome, which is critical for example, in choosing a course of treatment for a patient.

---

> > ### Author Response · Authors · 2021-11-19
> > **Response to reviewer r2Pt (2/3)**
> >
> > **Where does p_emp^i come from? How can one define it without giving reference to a sample-based estimation of the confidence score of f(x), which eventually boils down to the calibration we know? Having given a reference to a confidence score estimate, how is it different then from ordinary calibration?**
> >
> > $p_\text{emp}$ is an estimate of the conditional probability ${P}[y = 1| f(x) \in I( f(x_i) )]$ and $I( f(x_i) )$ is a small interval centered at $f(x_i)$. As explained in Section 3, if $f(x_i)$ is close to this value, then the model is well calibrated. We consider two approaches for estimating $p_\text{emp}^i$. (1) CaPE (bin) where we divide the training set into bins, select the bin $b_i$ containing $f(x_i)$ and set $p_{\text{emp}}^i=p_{\text{emp}}^{(b_i)}$ in equation 3. (2) CaPE (kernel) where $p_{\text{emp}}^i$ is estimated through a moving average with a kernel function (see Appendix E for more details). Both methods are efficiently computed by sorting the predictions $\hat p_i$.
> >
> > The reviewer is correct that the general methodology for estimating $p_\text{emp}$ is similar to what is used in evaluating calibration curves as a metric - we estimate the fraction of labels with network output $f(x)$ that have outcome 1, and use that to estimate the probability. Our contribution here is the way in which this estimate is used. In the calibration literature, $p_\text{emp}$ is used only for evaluation. Here, we incorporate $p_\text{emp}$ into our objective function, and use it to obtain better calibrated probability estimates.
> >
> > The reviewer makes an important point regarding the connection of calibration and probability estimation: At the end of the day, both in calibration and in probability estimation the goal is to quantify the uncertainty, so you can apply calibration methods for probability estimation. We completely agree with this, and we have edited the manuscript to make this clear. That said, to the best of our knowledge the majority of works that have proposed and evaluated these methods $\textbf{have focused exclusively on classification}$. We show through extensive experiments on synthetic data and three real-world datasets that these methods can also be effective for probability estimation (and in addition propose a novel method that outperforms them).
> >
> > **The "function kernel" inside Algorithm 1 appears to be the only part where one can argue about a methodological novelty. This function only replaces the Dirac measure used by standard binning with a kernel density estimator. This is a standard trick to improve numerical stability, but it is hard to say that this much makes a scientific novelty worthwhile to be published as a main-track conference paper.**
> >
> > We do not claim that kernel estimation is a contribution of the paper (see contributions in the introduction). In our novel algorithm for probability estimation we suggest to calibrate the estimation using the calibration curve estimate, we note that there are several ways to assign an empirical probability label $p_\text{emp}$ to a sample point. We show that the algorithm is robust with respect to a particular label assignment scheme by comparing the results of both binning and the kernel estimation.
> >
> > Regarding our methodological contributions, we propose a novel method for probability estimation, which exploits the early-learning phenomenon, whereby a model trained with cross-entropy produces well calibrated probability estimates up until a certain point, but then overfits the specific outcomes (see Figure 5). We leverage this phenomenon by continuing to train from that point by alternately minimizing a calibration cost function based on the model probability predictions, and a cross-entropy cost function. This enables the model to continue learning (see the validation curve in Figure 5).
> >
> >
> > **The paper lacks a clear focus in its core theme and a consistent logical flow. Is the main message how good Algorithm 1 is? Then the justification of the algorithm and explanation of why it works in the way it works is missing. Is it the allegedly new applications where uncertainty calibration is used?**
> >
> > We appreciate the reviewers’ comments on the clarity of the logical flow, which have inspired us to make some edits which have improved the manuscript. We set out to first establish the difference between probability estimation and classification problems, and based on that to investigate appropriate metrics, and possible solutions to the observed overfitting phenomenon. The proposed solution is novel in the sense that the calibration metric is used as a regularizer, drawing the network into a calibrated mode. As we show, it produces state of the art performance on real datasets with inherent uncertainty, as well as on synthetic datasets.

---

> > > ### Author Response · Authors · 2021-11-19
> > > **Response to reviewer r2Pt (3/3)**
> > >
> > > **References to the closest work are missing. I also do not think these are such novel applications of uncertainty calibration For instance see:
> > > Thagaard et al., Can you trust predictive uncertainty under real dataset shifts in digital pathology?, MICCAI 2020**
> > > Thank you for the reference, we have now cited it. Note that it targets a classical calibration problem where the objective is to classify correctly the present state of the patient under out of distribution images at the time of inference, rather than make predictions that are inherently uncertain. Our work is complementary as it addresses a future prognosis of the patient, trying to estimate the state the patient will be in several years, which is inherently uncertain. We have tried to be exhaustive in our references, but would appreciate any other suggestions of related works.

---

> > > > ### Comment · Reviewer_r2Pt · 2021-11-29
> > > > **Keep my score**
> > > >
> > > > Thanks for your detailed answer. I am afraid I am not convinced. The updated version also does not address some issues that look to me fundamental. The primary one is the claimed novelty of the very notion of probability estimation as a problem setup. I still do not see any difference of it from the reliability of the ordinary confidence score of a probabilistic classifier. The description in the paragraph "Probability Estimation" points only to the effect of heteroscedastic noise on calibration. However, I do not get why it should change anything in the way the existing approaches quantify uncertainty or why it should provide a conceptual advantage. An indicator that it does not is that the experiments report results on the classical reliability diagrams and calibration scores.

---

> > > > > ### Author Response · Authors · 2021-11-29
> > > > > **Clarification on our contributions**
> > > > >
> > > > > Thank you for your feedback. As we explained in our rebuttal, **there is no claimed novelty of the very notion of probability estimation as a problem setup**. We are merely making sure that the reader understands that we are targeting problems with inherent uncertainty. We do however propose **a novel method that outperforms state-of-the-art methods on three real-world datasets and several synthetic-data scenarios**. Taking that into account, we would really appreciate some feedback about our actual claimed contributions.

---

> > > > > > ### Comment · Reviewer_r2Pt · 2021-11-29
> > > > > > **Contributions**
> > > > > >
> > > > > > a) See paragraph 3 of my main review for my comments on the degree of methodological novelty of your work. It is simply a standard smoothing trick that can be applied to many things as a post-processing step.
> > > > > >
> > > > > > b) Regarding the novelty of the probability estimation, Fig 2 hints to a difference of the studied problem from calibration and I simply do not buy this argument. It is still calibration under label noise.
> > > > > >
> > > > > > In the updated version, three contributions are stated as contributions in the last paragraph of the Introduction. The first two are only data sets on which a standard calibration problem is studied (see my point (b)). If you claim something else, then you are also claiming novelty on the experiment setup, hence some new aspect of quantified uncertainty, contrarily to your last comment. The third claim is the kernel density estimator applied for smoothing, for which my conclusion is (a).

---

> > > > > > > ### Author Response · Authors · 2021-11-29
> > > > > > > **Clarification**
> > > > > > >
> > > > > > > Thank you for your rapid response. There are some aspects of it that are somewhat unclear to us:
> > > > > > >
> > > > > > > (1) When you refer to “a standard smoothing trick that can be applied to many things as a post-processing step” are you referring to our method? We propose a modification to the cost function used for training based on probability estimates driven by the early-learning phenomenon. The novelty is not the kernel density estimation, but rather using a calibration-based regularization term to avoid overfitting. We would really appreciate it if you could refer us to any previous publication that has proposed anything similar to this.
> > > > > > >
> > > > > > > (2) Figure 2 explains that a perfectly calibrated method can be really bad for probability estimation. If you do not “buy” this, please read footnote 1 at the bottom of the page.
> > > > > > >
> > > > > > > (3) Regarding our additional contributions apart from the novel method. We believe that it is crucial to test algorithms for probability estimation or calibration on problems with inherent uncertainty that reflect important real-world scenarios such as climate prediction, survival prediction, or collision prediction. We also believe that it is important to study what metrics are appropriate for this type of problem. We would again appreciate any references to works that have done this in the past.

---

### Official Review · Reviewer_QWNy · 2021-11-04

**Correctness:** 3
**Technical Novelty And Significance:** 3
**Empirical Novelty And Significance:** 3
**Recommendation:** 6
**Confidence:** 3

**Main Review:**

The idea is quite straightforward (which is not bad) and seems to be working well, as the authors compare their method to a number of baselines and show it is competitive. The idea is to a degree non-intuitive (at least to me a bit), but it does show good results. The writing is confusing though, and the authors could have done a much better job on that side. Please see details comments below.

**Summary Of The Paper:**

The authors propose a method to ensure calibration of probability outputs during training. They do so by adding an explicit penalty to push the output probability values towards an empirical estimate of the actual probability for the current inferred probability. They compare the method to a number of baseline approaches, showing the benefits on both synthetic and real-world data sets.

**Summary Of The Review:**

Detailed comments:
- "classification problems without inherent uncertainty", I am not sure that such problem exist. Please rephrase or elaborate further.
- Minor: make sure that all notation is correct (e.g., scalars are not bolded), both in the figures and in the text.
- "of just 75%", can you please clarify how?
- It is not fully clear why MSE_p is declared gold-standard, please clarify why.
- For Fig 4 results, more details are needed. E.g., early stopping is not well explained.
- I suggest moving Fig 5 much later, it seems introduced too early.
- For the kernel function, it is not clear whether the r-window is in the input or output space.
- In Section 6, uniform is not really uniform? Since it still increases with z.
- It is unclear which models were actually used for different data sets. This should be explained to a greater detail.
- In the last paragraph in Section 6, please explain better how each scenario aligns with the data sets, it is unclear as is. In fact, currently the entire section feels hand-wavy.
- Also, the last sentence doesn’t parse well, please rephrase.
- I recommend removing the last sentence in the Conclusion section.

---

> ### Author Response · Authors · 2021-11-19
> **Response to reviewer QWNy (1/2)**
>
> 1. **"classification problems without inherent uncertainty", I am not sure that such problem exist. Please rephrase or elaborate further**.
>
> In typical classification applications such as image recognition in benchmarks datasets like MNIST, CIFAR-100, or ImageNet, the label associated with each example is determined according to the content of the image. An image that contains a cat is labeled as “cat”. There is no uncertainty about whether there is a cat or not. In contrast, in probability-estimation problems, the label indicates an outcome that is **not fully determined by the image**. The three real-world applications we focus on in the paper provide specific examples. Given a certain histopathology image a cancer patient may or may not die in the next five years. Given a certain radar image it may or may not rain 30 minutes later. Given a certain dashboard image, there may or there may not be an accident a few seconds later. The specific outcome has inherent uncertainty because it depends on factors that are not captured in the data (for example, comorbidities or other health factors in cancer survival prediction, wind changes in weather prediction, and road conditions or driver expertise in collision prediction). However, the data does capture important information that makes it possible to determine the **probability of the outcome** given the observed data. We will add further clarification to ensure that this is clear in the paper.
> That said, the reviewer makes an interesting point that even in classification, there may be some uncertainty related to the labeling process. However, this is usually interpreted as label noise, which should be corrected, rather than as capturing inherent uncertainty in an outcome associated with the data. We mention this problem in the second paragraph of Section 4.
>
> 2. **Minor: make sure that all notation is correct (e.g., scalars are not bolded), both in the figures and in the text.**
>
> Thank you for pointing this out. We have corrected and standardized them in the updated version.
>
> 3.**"of just 75%", can you please clarify how?**
>
> We agree with the reviewer that more details were needed to make the explanation clear. We have added them to the paper. A perfect model (in terms of probability estimation), would assign 0.25 to the blue class and 0.75 to the red class. In order to maximize classification accuracy, we would predict 1 when the model outputs 0.75 (red examples) and 0 when it outputs 0.25 (blue examples). However, 25% of red examples have an outcome of 0, and 25% of blue examples have an outcome of 1. As a result, the model would only have 75% accuracy (it would be wrong 25% of the time).
>
> 4. **It is not fully clear why MSE_p is declared gold-standard, please clarify why.**
>
> In the case of simulated datasets, we have access to the latent ground truth probabilities. The actual outcomes are then sampled using these probabilities. In probability estimation, our goal is to estimate these ground-truth probabilities from the input data. It is therefore natural to use a metric that compares the ground-truth and estimated probabilities, and the l2 norm which corresponds to the mean-square error between them is a natural choice (although other distances could be possible).
>
> 5. **For Fig 4 results, more details are needed. E.g., early stopping is not well explained.**
>
> Thank you for pointing this out. We have updated the caption of Fig 4 to clarify different scenarios like “Infinite data”, and  “Early stopping”. By “Early stopping”, we mean a model which is early stopped based on the lowest validation cross-entropy loss.
>
> 6. **I suggest moving Fig 5 much later, it seems introduced too early.**
>
> The rationale of introducing Fig 5 in the Methods section is to explain and motivate our method, which prevents overfitting through the use of calibration loss. The figure shows that the method achieves this, further reducing the validation cross-entropy loss and improving performance. We have added additional clarification to explain the significance of the figure for that section.
>
> 7. **For the kernel function, it is not clear whether the r-window is in the input or output space.**
>
> The r-window is in the output space of probability. Thank you for pointing it out we have made it clearer in the updated version.
>
> 8. **In Section 6, uniform is not really uniform? Since it still increases with z.**
>
> Thank you for pointing this out. We agree the terminology is confusing. We have renamed it to “Linear” in the updated version.

---

> > ### Author Response · Authors · 2021-11-19
> > **Response to reviewer QWNy (2/2)**
> >
> > 9. **It is unclear which models were actually used for different data sets. This should be explained to a greater detail.**
> >
> > We briefly described the model architectures used for different datasets in Appendix H but we have further elaborated it to make it clearer in the updated version. For the Collision prediction dataset, we used a VGG model for the task of probability estimation. For the weather forecasting dataset, we used a ResNet-18 model. And for the cancer survival prediction dataset, first, we extracted representations of each patch of a whole-slide image using a model trained in a self-supervised manner (MoCo presented in (Chen et al., 2020)). The extracted representations of individual patches are, then, combined using a multiple-instance learning framework to predict the 0-1 survival outcome for the whole slide. For the simulated dataset, we use ResNet-18 for all our experiments.
> >
> > 10. **In the last paragraph in Section 6, please explain better how each scenario aligns with the data sets, it is unclear as is. In fact, currently the entire section feels hand-wavy.**
> >
> > Thank you for pointing this out. We realize that we do not clearly explain how different scenarios align with different datasets. To do so, we have added an appendix (Appendix K) which demonstrates the similarity between empirical probability curves of different real-world datasets and different scenarios of the simulated dataset. For the cancer survival dataset, the empirical probabilities are clustered in the center (0.4-0.6) similar to the “Centered” scenario. For the weather forecasting dataset, the probabilities are uniformly distributed across 0-1 similar to the “Uniform” (new name “Linear”) case. For the collision prediction dataset, the majority of the data points are clustered in the lower probability region which makes it similar to the “Skewed” scenario.
> >
> > 11. **Also, the last sentence doesn’t parse well, please rephrase.**
> >
> > Thanks, we have rephrased the sentence.
> >
> > 12. **I recommend removing the last sentence in the Conclusion section.**
> >
> > Thank you, we have removed the sentence.

---

> > > ### Comment · Reviewer_QWNy · 2021-11-30
> > > **Post-rebuttal response**
> > >
> > > I would like to thank the authors for the detailed response! It is highly appreciated.
> > >
> > > Having read the rebuttal and the other reviews, I would like to remain with the current score, and am potentially leaning toward weak rejection in the light of other reviewers' comments. I am also a bit confused by the discussion of a somewhat novel problem setting, as it seems that the current work already covers it. I do appreciate some of the novel ideas that the authors introduced (such as the new loss), but it is insufficient for a clear acceptance vote in the main conference in the light of the other discussion.

---

### Decision · Program_Chairs · 2022-01-20

**Decision:**

Reject

**Comment:**

The paper addresses the problem of uncertainty quantification in deep neural nets. The authors introduces the CaPE calibration loss to deal with the inherent uncertainty in probabilistic prediction, e.g. medical prognosis, weather prediction or collision prediction.

The paper initially received contrasted reviews: two weak acceptance, one weak rejection, and one strong rejection recommendation. The main limitation pointed out by reviewers related to the unclear definition of the problem setting, the limited contributions, and clarifications on experiments (comparison with deep ensembles). After authors' feedback, the reviewers were not convinced by the clarification on the problem setting, and there was a consensus among reviewers to reject the paper.

The AC's own readings confirmed the concerns raised by the reviewers, and also identifies additional shortcomings of the current submission. The paper addresses the problem of proper quantification of data uncertainty (generally referred as aleatoric uncertainty), and the CaPE calibration loss should be positioned with respect to the literature on the topic. The AC thus recommends rejection, but encourages the authors to re-submit their work after specifying the focus and motivation of their work.